# Gradient descent in matrix factorization: Understanding large initialization

Hengchao Chen[1]          Xin Chen[2]          Mohamad Elmasri[1]          Qiang Sun[1,3]

[1]University of Toronto
[2]Princeton University
[3]MBZUAI

## Abstract

Gradient Descent (GD) has been proven effective in solving various matrix factorization problems. However, its optimization behavior with large initial values remains less understood. To address this gap, this paper presents a novel theoretical framework for examining the convergence trajectory of GD with a large initialization. The framework is grounded in signal-to-noise ratio concepts and inductive arguments. The results uncover an implicit incremental learning phenomenon in GD and offer a deeper understanding of its performance in large initialization scenarios.

## 1 INTRODUCTION

Low-rank optimization is pivotal in various applications, including matrix completion [Candes and Recht, 2012], phase retrieval [Shechtman et al., 2015], matrix sensing [Recht et al., 2010], and others. One common approach to addressing low-rank problems is through convex relaxation techniques such as nuclear norm regularization [Recht et al., 2010]. While these methods offer robust statistical performance guarantees, they often incur substantial computational costs, potentially scaling cubically with matrix size [Chi et al., 2019]. In response, recent studies [Chen et al., 2019, Ma et al., 2018, Chi et al., 2019] have proposed employing matrix factorization, which naturally encodes low rankness and significantly reduces the computational expenses per iteration when using gradient descent. However, a primary concern remains: the non-convex nature of the resulting objective necessitates a thorough investigation into the convergence properties of gradient descent (GD).

Recent research has demonstrated that GD effectively solves a variety of low-rank problems. Ma et al. [2018] show that

with a benign initialization[1], GD converges linearly to the global minima in applications such as matrix completion, phase retrieval, and blind deconvolution. Similarly, Zhu et al. [2021] explore the optimization landscape of matrix sensing, finding that GD, when starting with a benign initialization, converges linearly to the global minima. These findings highlight the intriguing optimization properties of GD in non-convex settings and have catalyzed many studies in low-rank matrix optimization [Chi et al., 2019].

Unlike in convex optimization, initialization is critical in non-convex optimization. The aforementioned works establish global convergence under the assumption of a benign initialization, necessitating carefully designed starting points. This prompts a critical question: Is random initialization[2] sufficient to ensure fast global convergence? For certain problems, the answer is affirmative. Chen et al. [2019] show that with a random initialization, GD quickly converges globally in phase retrieval. Stöger and Soltanolkotabi [2021] show that GD with a small random initialization[3] also achieves rapid global convergence in matrix sensing. Subsequent studies have further examined GD with small random initialization in matrix sensing, addressing different aspects such as the asymmetric case [Jiang et al., 2023] and the incremental learning phenomenon [Jin et al., 2023]. These investigations, except Chen et al. [2019], all assume a small initialization, which we discuss below in detail.

In this paper, we investigate the convergence properties of GD with a large initialization. Using a large initialization often helps reduce the training time, and is widely adopted in neural network training [Sun, 2020]. For simplicity, we limit our analysis to a population version[4] of the matrix sensing

---

[1]A benign initialization refers to starting the algorithm near the global minima.

[2]Random initialization selects an initial point randomly, typically far from the global minima.

[3]Small or large initialization refers to whether the norm of the initial point is near zero or substantially larger, respectively.

[4]A brief introduction to the matrix sensing problem can be found in Appendix A. The difference between matrix sensing and

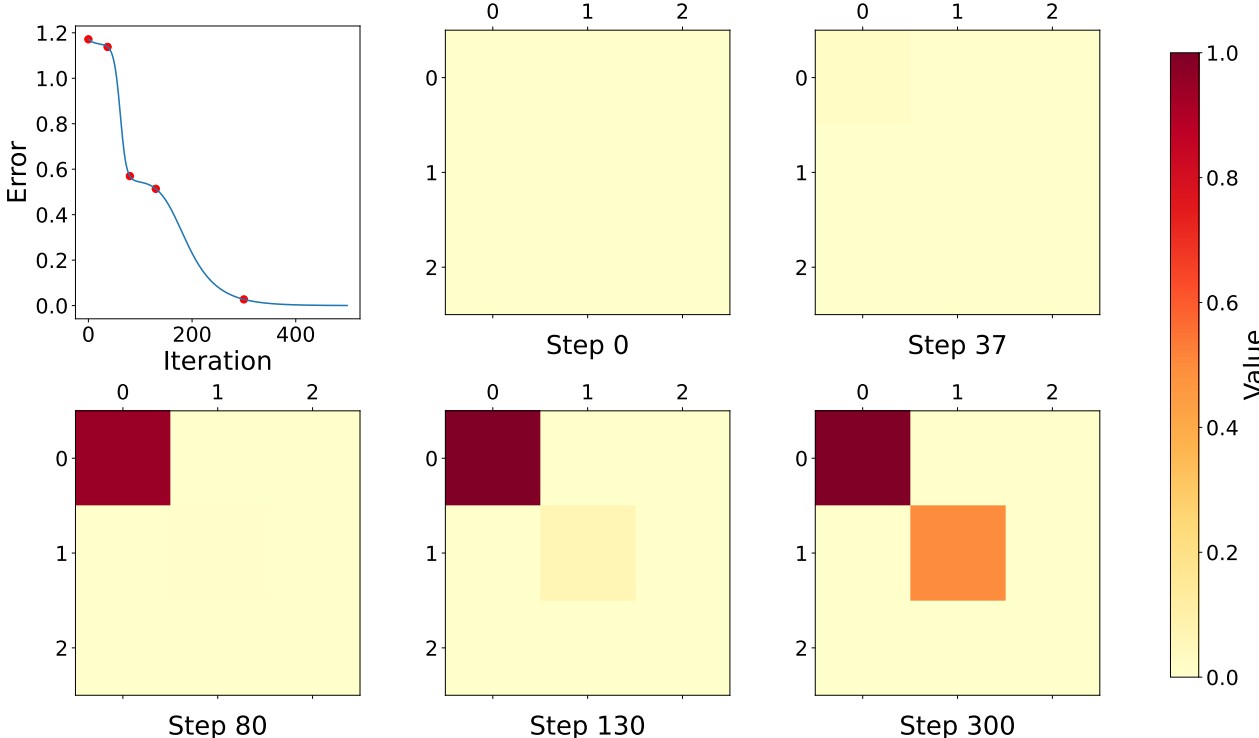

Figure 1: The top left panel shows the errors $\|\mathbf{\Sigma}_r - \mathbf{X}_t\mathbf{X}_t^\top\|_\mathrm{F}$ over iterations. The rest panels show the heat maps of the top three rows and columns of $\mathbf{X}_t\mathbf{X}_t^\top$ at iterations $t = 0, 37, 80, 140$, and $300$, corresponding to the red points in the top left panel.

problem. Specifically, we aim to study the convergence properties of GD when solving the following symmetric matrix factorization problem:

$$\mathbf{X}^* = \arg\min_{\mathbf{X}\in\mathbb{R}^{d\times r}} \|\mathbf{\Sigma} - \mathbf{X}\mathbf{X}^\top\|_\mathrm{F}^2, \qquad (1)$$

where $\mathbf{\Sigma} \in \mathbb{R}^{d\times d}$ is a positive semi-definite matrix of rank at least $r$. While it is straightforward to show that $\mathbf{X}^*\mathbf{X}^{*\top} = \mathbf{\Sigma}_r$, with $\mathbf{\Sigma}_r$ being the best rank-$r$ approximation of $\mathbf{\Sigma}$, establishing the global convergence theory of GD in this problem is non-trivial. Let the initialization be:

$$\mathbf{X}_0 = \varpi\mathbf{N}_0 \in \mathbb{R}^{d\times r}, \text{ entries of } \mathbf{N}_0 \text{ being i.i.d. } \mathcal{N}(0, 1/d).$$

Let $\kappa = \lambda_1/\Delta \geq 1$ be the conditional number, with $\lambda_1$ being the largest eigenvalue of $\mathbf{\Sigma}$ and $\Delta$ denoting the eigengap. Prior works typically assume a small initialization, aka $\varpi = d^{-\iota(\kappa)}$ for some positive increasing function $\iota(\cdot)$[5]. For instance, Stöger and Soltanolkotabi [2021] require

$$\varpi \lesssim d^{-3\kappa^2}, \qquad (2)$$

its population version will also be discussed.

[5]We call $\mathbf{X}_0$ a small initialization when $\varpi$ tends to zero as $d$ increases. This is because a standard statistical analysis shows that $\|\mathbf{X}_0\|_\mathrm{F} = \Theta(\varpi)$, which omits constants that are independent of $d$.

and Jin et al. [2023] need a even smaller $\varpi$. Such $\varpi$ decays rapidly to zero as $d$ or $\kappa$ increases. In our study, we do not impose such conditions and only require $\varpi = O(1)$. We show that this requirement is necessary since GD will diverge when $\varpi$ is larger than this order. We refer to an initialization with $\varpi = O(1)$ as a large initialization, reflecting its relative magnitude in comparison to typical values considered in the field. Notably, our theories are applicable to the small initialization scenario as well, which is a special case of $\varpi = O(1)$.

## 1.1 OUR CONTRIBUTION

Our main result is outlined in the informal theorem below, with its rigorous counterparts collected later. This result is established through a novel signal-to-noise ratio (SNR) based approach, combined with an inductive argument.

**Theorem 1 (Informal)** *Assume that $\mathbf{\Sigma}$ is a positive semi-definite matrix with its top $r + 1$ eigenvalues being distinct and arranged in descending order. Let $\mathbf{X}_t$ be the GD sequence for problem (1) with $\mathbf{X}_0 = \varpi\mathbf{N}_0$, where $\varpi$ is a positive constant independent of $d$ and $\mathbf{N}_0 \in \mathbb{R}^{d\times r}$ has independent $\mathcal{N}(0, 1/d)$ entries. Then*

*1. The GD sequence converges to the global minima almost*

*surely [Lee et al., 2016, Zhu et al., 2021];*

2. *A comprehensive trajectory analysis of GD is provided, indicating that eigenvectors associated with larger eigenvalues are learned first;*

3. *Under an additional assumption, GD achieves $\epsilon$-accuracy in $\mathcal{O}(\log(1/\epsilon) + \log(d))$ steps.*

Property 1 is a direct consequence of prior works. It guarantees that our trajectory analysis is valid almost surely. Property 2 shows that the top eigenvectors can be learned first, unaffected by the behavior of later signals. This point will be clarified later. Property 3 describes the fast global convergence of GD under an additional assumption concerning the saddle point escaping property. The verification of this assumption requires new theoretical techniques, which we defer to future research.

Now we elucidate Property 2 and our theoretical analysis through a simple yet representative example concerning rank-two matrix approximation. The experiment is conducted with parameters: $d = 4000$, $r = 2$, and $\boldsymbol{\Sigma} = \mathrm{diag}(1, 0.5, \boldsymbol{e})$, where $\boldsymbol{e} \in \mathbb{R}^{d-r}$ forms an arithmetic sequence decreasing from 0.3 to 0. The initialization matrix is set to $\boldsymbol{X}_0 = 0.5\boldsymbol{N}_0$ with entries of $\boldsymbol{N}_0$ independently sampled from $\mathcal{N}(0, \frac{1}{d})$. The GD iterations $\boldsymbol{X}_t$ are computed using a step size of 0.1. We evaluate errors using $\|\boldsymbol{\Sigma}_r - \boldsymbol{X}_t\boldsymbol{X}_t^\top\|_{\mathrm{F}}$ with $\boldsymbol{\Sigma}_r = \mathrm{diag}(1, 0.5, 0, \ldots, 0)$ representing the best rank-$r$ matrix approximation to $\boldsymbol{\Sigma}$. In Figure 1, the error trajectory is plotted, highlighting key inflection points and featuring heat maps of the first three rows and columns of $\boldsymbol{X}_t\boldsymbol{X}_t^\top$ at these steps. Observations indicate that GD undergoes an incremental learning process, characterized by error curves that exhibit both flat and steep segments.

To interpret the error trajectory in Figure 1, we analyze the first $r$ rows of $\boldsymbol{X}_t$ individually. Specifically, we examine the dynamics of the quantities $\sigma_1(\boldsymbol{u}_{k,t})$ and $\sigma_1(\boldsymbol{u}_{k,t}\boldsymbol{K}_{k,t}^\top)$, where $\boldsymbol{u}_{k,t}$ is the $k$-th row of $\boldsymbol{X}_t$ and $\boldsymbol{K}_{k,t}$ is the $(k+1)$-to-$d$-th rows of $\boldsymbol{X}_t$. These quantities correspond to the diagonal and off-diagonal elements in the heat map of $\boldsymbol{X}_t\boldsymbol{X}_t^\top$. Hence, our mathematical analysis directly explains the dynamics observed in the heat maps of Figure 1. Our analysis on the SNR $\sigma_1^2(\boldsymbol{u}_{k,t})/\sigma_1(\boldsymbol{u}_{k,t}\boldsymbol{K}_{k,t}^\top)$ reveals that the off-diagonal elements decrease at a geometric rate once the signal strength $\sigma_1^2(\boldsymbol{u}_{k,t})$ reaches a specific threshold. This observation motivates us to employ an inductive argument to analyze the whole convergence trajectory.

Lastly, we highlight an ancillary contribution regarding the convergence of GD with a benign initialization. Prior works, such as Zhu et al. [2021], only prove the global linear convergence of GD when $\boldsymbol{\Sigma}$ is exactly of rank $r$. In contrast, our work extends these results by proving global linear convergence of GD for all matrix approximation problems, allowing for $\boldsymbol{\Sigma}$ whose rank exceeds $r$. Our analysis is based

on an SNR argument, which distinguishes from the prior landscape-based analysis. Additionally, this analysis can be further applied to offer a new proof for the global linear convergence of benignly initialized GD in matrix sensing problems.

## 1.2 RELATED WORK

Matrix factorization based low rank matrix optimization has received significant attention in recent years [Chi et al., 2019]. A primary challenge involves analyzing the optimization properties. Previous studies have approached this issue through various angles, including examing the optimization landscape [Sun et al., 2016, 2018, Zhu et al., 2021] and directly conducting convergence analyses [Ma et al., 2018, Chen et al., 2019, Stöger and Soltanolkotabi, 2021]. Lee et al. [2016] show that GD escapes saddle points almost surely under the strict saddle point condition, implying global convergence in scenarios where all local minima are also global minima, and all saddle points are strict.

In our paper, we refer to the implicit incremental learning phenomenon as the prioritized learning of the top eigenstructure. This phenomenon has been investigated from different perspectives. Li et al. [2020] show that in matrix factorization, gradient flow with infinitesimal initialization is mathematically equivalent to greedy low-rank learning under specific assumptions. Jin et al. [2023] show that in matrix sensing, GD with a small initialization exhibits an incremental learning phenomenon. Simon et al. [2023] observe that a similar incremental learning phenomenon exists in self-supervised learning when a small initialization is employed. However, none of these works explore the large initialization regime, which we will investigate for the first time in this paper.

## 1.3 PAPER OVERVIEW

The rest of this paper proceeds as follows. Section 2.1 reviews the usage of SNR analysis for rank-one matrix approximation. Section 2.2 uses the SNR analysis to prove the local linear convergence of GD in general rank problems. In Section 3, we examine the random initialization. Specifically, Section 3.1 reviews small initialization while Section 3.2 considers large initialization and presents our main result. A sketch of proof is provided in Section 4. Concluding discussions are presented in Section 5 and all detailed proofs are collected in the Appendix.

## 2 A WARM-UP

In this section, we first introduce the SNR analysis for rank-one matrix approximation problems as developed by Chen et al. [2019]. We then discuss the challenges involved in

extending this analysis to the case of general rank. Despite these challenges, it is possible to extend the analysis with the use of a benign initialization. We provide such an extension in Theorem 2, which generalizes the previous results.

## 2.1 RANK-ONE MATRIX APPROXIMATION

We begin with a review of rank-one matrix approximation. In this context, Chen et al. [2019] demonstrate that GD with a large random initialization exhibits linear convergence to the global minima by leveraging an SNR argument. Specifically, consider problem (1) with $r = 1$ and assume[6] $\boldsymbol{\Sigma} = \mathrm{diag}(\lambda_1, \ldots, \lambda_d)$ is diagonal with decreasing diagonal elements and $\lambda_1 > \lambda_2$. Let the initialization vector $\boldsymbol{x}_0 \in \mathbb{R}^d$ be such that its first entry is non-zero and its norm $\|\boldsymbol{x}_0\|$ is less than $2\lambda_1$. Then $\boldsymbol{x}_t \boldsymbol{x}_t^\top$ rapidly converges to $\mathrm{diag}(1, 0, \ldots, 0)$. The vector $\boldsymbol{x}_t$ is updated according to the GD rule:

$$\boldsymbol{x}_t = \boldsymbol{x}_{t-1} + \eta(\boldsymbol{\Sigma} - \boldsymbol{x}_{t-1}\boldsymbol{x}_{t-1}^\top)\boldsymbol{x}_{t-1}, \qquad (3)$$

where $\eta$ is the learning rate.

Chen et al. [2019] first decompose $\boldsymbol{x}_t$ as $\boldsymbol{x}_t = (a_t, \boldsymbol{b}_t)^\top$, where $a_t \in \mathbb{R}$ and $\boldsymbol{b}_t \in \mathbb{R}^{d-1}$. The GD rule is then rewritten as

$$a_t = a_{t-1} + \eta\lambda_1 a_{t-1} - \eta(a_{t-1}^2 + \|\boldsymbol{b}_{t-1}\|^2)a_{t-1}, \qquad (4)$$

$$\boldsymbol{b}_t = \boldsymbol{b}_{t-1} + \eta\boldsymbol{\Sigma}_{\mathrm{res}}\boldsymbol{b}_{t-1} - \eta(a_{t-1}^2 + \|\boldsymbol{b}_{t-1}\|^2)\boldsymbol{b}_{t-1}, \quad (5)$$

where $\boldsymbol{\Sigma}_{\mathrm{res}} = \mathrm{diag}(\lambda_2, \ldots, \lambda_d)$. Let $\alpha_t = |a_t|$ and $\beta_t = \|\boldsymbol{b}_t\|$, and assume $\eta\lambda_1$ is smaller than some constant, say $\frac{1}{12}$. Then we have

$$\alpha_t = (1 + \eta\lambda_1 - \eta\alpha_{t-1}^2 - \eta\beta_{t-1}^2)\alpha_{t-1}, \qquad (6)$$

$$\beta_t \le (1 + \eta\lambda_2 - \eta\alpha_{t-1}^2 - \eta\beta_{t-1}^2)\beta_{t-1}. \qquad (7)$$

By dividing (7) by (6), it follows that

$$\frac{\beta_t}{\alpha_t} \le \frac{1 + \eta\lambda_2 - \eta\alpha_{t-1}^2 - \eta\beta_{t-1}^2}{1 + \eta\lambda_1 - \eta\alpha_{t-1}^2 - \eta\beta_{t-1}^2} \cdot \frac{\beta_{t-1}}{\alpha_{t-1}}$$

$$\le (1 - \frac{\eta\Delta}{3}) \cdot \frac{\beta_{t-1}}{\alpha_{t-1}}, \qquad (8)$$

where $\Delta = \lambda_1 - \lambda_2$ is the eigengap and the second inequality uses the fact that for all $s \in [-1/2, 1/2]$,

$$h(s) = \frac{1 - \eta\Delta/2 + s}{1 + \eta\Delta/2 + s} \le h(1/2) \le 1 - \frac{\eta\Delta}{3}. \qquad (9)$$

Inequality (8) indicates that the ratio $\beta_t/\alpha_t$ decreases to zero geometrically fast. Using this result, Chen et al. [2019] establish that $\beta_t$ and $\alpha_t$ rapidly converge to zero and $\lambda_1$, respectively. In our paper, we refer to this argument as an SNR analysis, designating $\alpha_t$ as the signal strength and $\beta_t$ as the noise strength.

---

[6]There is no loss of generality in assuming that $\boldsymbol{\Sigma}$ is diagonal, as the analysis of GD is invariant to orthogonal rotations. For a detailed explanation, please refer to Appendix A.2.

## 2.2 THE GENERAL RANK CASE: CHALLENGES AND A SOLUTION

Generalizing the SNR argument to general rank problems introduces significant challenges. For example, the global minima cannot simply be characterized by the two real numbers $\alpha_t$ and $\beta_t$. Identifying other effective measures that characterize the GD sequence, and providing a dynamic analysis akin to that in equations (6) and (7), is notably difficult. Fundamentally, this difficulty arises from the heterogeneity across different dimensions or, more formally, from the non-commutativity of matrix multiplication.

One way to address this challenge involves using a benign initialization with a high initial SNR. This strategy facilitates the extension of the SNR analysis to general rank problems and enables the establishment of the local linear convergence for GD. Consider problem (1) with a general $r$ and assume $\boldsymbol{\Sigma} = \mathrm{diag}(\lambda_1, \ldots, \lambda_d)$ is diagonal with decreasing diagonal elements, where the eigengap $\Delta := \lambda_r - \lambda_{r+1} > 0$. Let $\boldsymbol{X}_0 \in \mathbb{R}^{d \times r}$ be an initialization point. Then the GD update rule is:

$$\boldsymbol{X}_t = \boldsymbol{X}_{t-1} + \eta(\boldsymbol{\Sigma} - \boldsymbol{X}_{t-1}\boldsymbol{X}_{t-1}^\top)\boldsymbol{X}_{t-1}, \qquad (10)$$

where $\eta$ is the learning rate.

For the SNR argument, we decompose $\boldsymbol{X}_t$ into $(\boldsymbol{U}_t^\top, \boldsymbol{J}_t^\top)^\top$, where $\boldsymbol{U}_t$ consists of the first $r$ rows of $\boldsymbol{X}_t$ and $\boldsymbol{J}_t$ includes the remaining $d - r$ rows. Analogous to the rank-one case, $\boldsymbol{U}_t$ is considered the signal, being non-zero at the global minima, while $\boldsymbol{J}_t$ is considered the noise, being zero at the same. Adopting a benign initialization means that $\sigma_r(\boldsymbol{U}_0)$ is large and $\sigma_1(\boldsymbol{J}_0)$ is small. More precisely, we define the set $\mathcal{R}$ as:

$$\mathcal{R} = \left\{ \boldsymbol{X} = \begin{pmatrix} \boldsymbol{U} \\ \boldsymbol{J} \end{pmatrix} \mid \sigma_1^2(\boldsymbol{X}) \le 2\lambda_1, \sigma_r^2(\boldsymbol{U}) \ge \Delta/4, \right.$$

$$\left. \sigma_1^2(\boldsymbol{J}) \le \lambda_r - \Delta/2 \right\}. \qquad (11)$$

This set contains all the global minima of problem (1). Moreover, the SNR $\sigma_r^2(\boldsymbol{U})/\sigma_1^2(\boldsymbol{J})$ is larger than the constant $\Delta/(4\lambda_1)$ for any $\boldsymbol{X}$ in $\mathcal{R}$. In the Appendix, we demonstrate that if GD is initialized within $\mathcal{R}$, the sequence $\boldsymbol{X}_t$ will remain in $\mathcal{R}$ and the SNR will rapidly increase towards infinity. Consequently, we can establish the local linear convergence of GD as in Theorem 2, which is instrumental in analyzing random initialization. Later, we will show that, for any initialization point $\boldsymbol{X}_0 \notin \mathcal{R}$, the convergence of GD consists of two phases: the first phase, where the sequence enters $\mathcal{R}$, followed by the final global convergence phase. With Theorem 2, we only need to analyze the first phase.

**Theorem 2** *Suppose $\eta \le \frac{\Delta^2}{36\lambda_1^3}$, $\boldsymbol{X}_0 \in \mathcal{R}$, and $\boldsymbol{X}_t$ is the GD sequence given by (10). Then, for any small $\epsilon > 0$, we have $\|\boldsymbol{\Sigma}_r - \boldsymbol{X}_t\boldsymbol{X}_t^\top\| \le \epsilon$ in $\mathcal{O}(\frac{6}{\eta\Delta} \ln \frac{200r\lambda_1^3}{\eta\Delta^2\epsilon})$ iterations, where $\boldsymbol{\Sigma}_r = \mathrm{diag}(\lambda_1, \ldots, \lambda_r, 0, \ldots, 0)$.*

Prior studies on local linear convergence have either focused on the rank-one scenario [Chen et al., 2019] or assumed that $\Sigma$ is exactly of rank $r$ [Zhu et al., 2021], and their proof arguments do not directly apply to Theorem 2. In contrast, by utilizing an SNR argument, we establish local linear convergence for general cases. Our SNR analysis hinges on establishing a lower bound for the signal, $\sigma_r^2(U_{t+1})$, and an upper bound for the noise, $\sigma_1^2(J_{t+1})$. These bounds must be precisely related to facilitate the analysis of the ratio between $\mathrm{SNR}_{t+1}$ and $\mathrm{SNR}_t$, which poses significant challenges.

Moreover, while we assume $\Sigma$ is positive semi-definite for simplicity, our proof can easily be adapted to general symmetric case. It can also be modified to establish the local linear convergence of GD in matrix sensing scenarios [Zhu et al., 2021].

# 3 RANDOM INITIALIZATION

Benign initialization, while conceptually valuable, has limited practical utility because it often requires oracle information. This limitation is particularly pronounced in matrix sensing problems, where $\Sigma$ is observed only through random measurements [Stöger and Soltanolkotabi, 2021]. Consequently, researchers have shifted their focus towards random initialization. According to Theorem 2, the convergence analysis of GD simplifies to determining the duration required for the sequence to enter the set $\mathcal{R}$. Once within $\mathcal{R}$, the sequence is guaranteed to converge to the global minima exponentially fast.

## 3.1 SMALL RANDOM INITIALIZATION

Existing research, with the exception of the rank-one case, predominantly focuses on small random initialization. This approach assumes $X_0 = \varpi N_0$, where $N_0 \in \mathbb{R}^{d \times r}$ has independent $\mathcal{N}(0, \frac{1}{d})$ entries and $\varpi$ is notably small. Concentration arguments indicate the norm $\|X_0\|$ is of order $\mathcal{O}(\varpi)$. When $\varpi$ is sufficiently small, the higher-order term $X_.X_.^\top X_.$ in (10) becomes negligible in the early iterations. Consequently, during these early iterations, the GD iteration behaves like a power method:

$$X_t \approx X_{t-1} + \eta \Sigma X_{t-1}. \tag{12}$$

The eigenvectors associated with larger eigenvalues will be learned faster. Using the same $U, J$ notation as in Section 2.2, we obtain that $\sigma_r(U_{t+1})/\sigma_r(U_t)$ is greater than $\sigma_1(J_{t+1})/\sigma_1(J_t)$ for small $t$, indicating that the signal strength increases faster than the noise strength. By picking a sufficiently small $\varpi$, we can show that after $\mathcal{O}(\log(d))$ rounds, $\sigma_r^2(U_t)$ will rise above $\Delta/4$ while $\sigma_1(J_t)$ remains negligible. This rapid entry of the sequence $X_t$ into the region $\mathcal{R}$ facilitates global linear convergence, as shown by Stöger and Soltanolkotabi [2021]. Additionally, Jin et al.

[2023] explore the incremental learning behavior of GD under small $\varpi$ conditions. Other studies, such as those by Ma et al. [2022] and Soltanolkotabi et al. [2023], also examine GD with small initialization.

## 3.2 LARGE RANDOM INITIALIZATION

In practice, however, a large initialization is often employed, where $X_0 = \varpi N_0$ with $\varpi$ being a constant **independent of** $d$. In such cases, the arguments in Section 2.2 or Section 3.1 are inadequate. Specifically, the initial SNR is too low for the arguments in Section 2.2 to be applicable, and the initial magnitude $\|X_0\|$ is too high to use the arguments in Section 3.1. Thus, to examine large initialization, we need a more delicate analysis, as presented in this section.

Our analysis builds upon the discussions presented in Section 1.1. Specifically, we consider problem (1) with a general rank parameter $r$ and assume without loss of generality that $\Sigma = \mathrm{diag}(\lambda_1, \ldots, \lambda_d)$ is diagonal with decreasing diagonal elements. Suppose the leading $r + 1$ eigenvalues of $\Sigma$ are strictly decreasing and let $\Delta = \min_{i \leq r}\{\lambda_i - \lambda_{i+1}\} > 0$ be the eigengap. Our goal is to characterize the convergence of the GD sequence $X_t$ as defined in equation (10).

Following the discussion in Section 1.1, we define $u_{k,t}$ as the $k$-th row of $X_t$ and $K_{k,t}$ as the $(k + 1)$-to-$d$-th rows of $X_t$. The relationships between these definitions and the visualizations in Figure 1 are introduced in Section 1.1. Let

$$\mathcal{S} = \{X \in \mathbb{R}^{d \times r} \mid \sigma_1^2(X) \leq 2\lambda_1,$$
$$\sigma_1^2(K_k) \leq \lambda_k - \frac{3\Delta}{4}, \forall k \leq r\}, \tag{13}$$

be a subset of $\mathbb{R}^{d \times r}$ where $X$ and $K_k$, the $(k + 1)$-to-$d$ rows of $X$, both have bounded norms. We will show that $X_t$ quickly enters the set $\mathcal{S}$. The duration until this entry occurs is denoted by:

$$t_{\mathrm{init},1} = \min\{t \geq 0 \mid X_t \in \mathcal{S}\}.$$

To streamline our presentation, we introduce two constants $t^*$ and $t^\sharp$:

$$t^* = \log\left(\frac{\Delta^2}{8\lambda_1^3 + 144r^2\lambda_1}\right) / \log(1 - \eta\Delta/6), \tag{14}$$

$$t^\sharp = \log\left(\Delta/(4r)\right) / \log(1 - \eta\Delta/6). \tag{15}$$

Subsequently, we define the quantities

$$\{T_{u_k}, t_k, t_k^*, t_{\mathrm{init},k+1}\}$$

in a successive manner up to $T_{u_r}$:

- Define $T_{u_k}$, counted from $t_{\mathrm{init},k} + 1$, as the earliest time when the strength of the $k$-th signal, $\sigma_1^2(u_{k,t})$, first surpasses $\Delta/2$:

$$T_{u_k} = \min\{t \geq 0 \mid \sigma_1^2(u_{k,\,t+t_{\mathrm{init},k}}) \geq \Delta/2\};$$

- $t_k = t_{\text{init},k} + T_{\boldsymbol{u}_k} + t^*$;

- $t_k^*$ is the smallest integer for which the following inequality holds, indicating that the $(k+1)$-th signal strength no longer falls below a geometrically decaying sequence

$$r(1 - \eta\Delta/6)^{t_k^*}$$
$$\leq \sqrt{\frac{\Delta}{8}} \min\left\{\sigma_1(\boldsymbol{u}_{k+1,t_k+t_k^*}), \sqrt{\frac{\Delta}{2}}\right\}; \quad (16)$$

- $t_{\text{init},k+1} = t_k + t_k^*$.

These quantities are instrumental in characterizing the convergence of GD, and our primary objective is to upper bound these quantities. We first find that $t_k^* < \infty, \forall k \leq r$ almost surely when random initialization is utilized. This finding, articulated in Proposition 3, is supported by the theory of Lee et al. [2016] and the landscape analysis of Zhu et al. [2021]. A detailed proof of this result is deferred to the Appendix.

**Proposition 3** *Let $\eta \leq \frac{\Delta}{100\lambda_1^2}$ and $\boldsymbol{X}_t$ be the GD sequence initialized with $\boldsymbol{X} \in \mathbb{R}^{d \times r}$. Then the following set*

$$\{\boldsymbol{X} \in \mathbb{R}^{d \times r} | \sigma_1(\boldsymbol{X}) \leq 1/\sqrt{3\eta}, t_k^* = \infty \text{ for some } k \leq r\}$$

*is of Lebesgue measure zero.*

Motivated by this proposition, we introduce the following assumption and then present our main theorem. While our theorem is formulated under deterministic initialization, it remains applicable to scenarios involving random initialization.

**Assumption 4** *Assume that $t_k^* < \infty$ for all $k \leq r$.*

**Theorem 5** *Suppose $\eta \leq \frac{\Delta^2}{100\lambda_1^3}$, $\sigma_1(\boldsymbol{X}_0) \leq 1/\sqrt{3\eta}$, $\boldsymbol{X}_t$ is the GD sequence, and Assumption 4 holds. Then we have*

*(1) $\boldsymbol{X}_t \in \mathcal{R}$ for all $t \geq t_{\mathcal{R}} := t_{\text{init},r} + T_{\boldsymbol{u}_r} + t^* + t^\sharp$, where*

$$t_{\text{init},1} = \mathcal{O}\left(\frac{1}{\eta\lambda_1}\log\frac{1}{6\eta\lambda_1}\right) + \mathcal{O}\left(\frac{1}{\eta\Delta}\log\frac{8\lambda_1}{\Delta}\right),$$
$$T_{\boldsymbol{u}_k} = \mathcal{O}\left(\frac{4}{\eta\Delta}\log\frac{\Delta}{2\sigma_1^2(\boldsymbol{u}_{k,t_{\text{init},k}})}\right), \quad \forall k \leq r.$$

*(2) GD achieves $\epsilon$-accuracy, i.e., $\|\boldsymbol{\Sigma}_r - \boldsymbol{X}_t\boldsymbol{X}_t^\top\|_{\text{F}} \leq \epsilon$, in*

$$t_{\mathcal{R}} + \mathcal{O}\left(\frac{6}{\eta\Delta}\ln\frac{200r\lambda_1^3}{\eta\Delta^2\epsilon}\right) \quad (17)$$

*iterations.*

*(3) For all $k < r$ and $t \geq t_k$, both $\sigma_1(\boldsymbol{u}_{k,t}\boldsymbol{K}_{k,t}^\top)$ and $p_{k,t}$ converge to zero linearly fast:*

$$\sigma_1(\boldsymbol{u}_{k,t}\boldsymbol{K}_{k,t}^\top) \leq (1 - \eta\Delta/6)^{t-t_k},$$
$$|p_{k,t}| \leq (2\lambda_1 + \frac{24r}{\eta\Delta}) \cdot (1 - \eta\Delta/8)^{t-t_k}, \quad (18)$$

*where $p_{k,t} = \lambda_k - \sigma_1^2(\boldsymbol{u}_{k,t})$. This reveals the implicit incremental learning of GD.*

Theorem 5 imposes relatively mild conditions. First, the condition that $\sigma_1(\boldsymbol{X}_0) \leq 1/\sqrt{3\eta}$ holds with high probability when we pick $\boldsymbol{X}_0 = \varpi\boldsymbol{N}_0$ with $\varpi \lesssim 1/\sqrt{\eta}$, using the same $\boldsymbol{N}_0$ as previously discussed. This order is maximal, as the GD sequence may diverge when $\sigma_1(\boldsymbol{X}_0)$ exceeds this order. For example, consider $\boldsymbol{\Sigma} = \boldsymbol{0}$ and $\eta\sigma_1^2(\boldsymbol{X}_0) \geq 3$. By employing an inductive argument for the GD iteration (10), we can show that

$$\sigma_1(\boldsymbol{X}_{t+1}) \geq (\eta\sigma_1^2(\boldsymbol{X}_t) - 1) \cdot \sigma_1(\boldsymbol{X}_t) > 2\sigma_1(\boldsymbol{X}_t), \quad \forall t.$$

This result implies that the GD sequence diverges when $\varpi$ is a large constant. Consequently, it establishes the maximal order of $\varpi$ for convergence is $\mathcal{O}(1/\sqrt{\eta})$. The only possible improvement could be a constant factor. Additionally, this rate is independent of the dimension $d$, in stark contrast to the condition in the small initialization scenario (2) where $\varpi$ decays to zero exponentially fast as $d$ increases. This relaxed assumption makes our theorem applicable to large initialization. Second, Assumption 4 is considered mild as demonstrated in Proposition 3. Therefore, Theorem 5 is applicable across a wide range of contexts.

The conclusions of Theorem 5 are threefold. First, we upper bound all quantities except $t_k^*$ by logarithmic terms. These bounds partially explain the fast convergence of GD in Figure 1. Next, by combining property (1) with Theorem 2, we obtain the global convergence rate in (17). Third, we show that the $k$-th signal strength converges to the target value exponentially fast following the $t_k$th step. Crucially, this convergence is independent of $t_j^*$ for all $j \geq k$. This indicates that the convergence of the $k$-th signal is not affected by the behavior of subsequent signals ($(k+1)$-to-$r$-th), exemplifying an implicit incremental learning phenomenon in GD.

Finally, if we make an additional assumption, we can obtain the fast global convergence of GD in Theorem 7.

**Assumption 6** *Assume $t_k^* = \mathcal{O}(\log(d))$ for all $k \leq r$.*

**Theorem 7** *Assume that conditions in Theorem 5 and Assumption 6 hold. Then GD achieves $\epsilon$-accuracy in $\mathcal{O}(\log(d) + \log(1/\epsilon))$ iterations.*

Assumption 6 is particularly nuanced as it upper bounds the quantity $t_k^*$. We call it a transition assumption because

it facilitates the analytical progression from the analysis of the $k$-th row to the $(k + 1)$-th row, positing the transition time is $\mathcal{O}(\log(d))$. Verifying this assumption is challenging and we leave it to the future work.

# 4 PROOF SKETCH

In this section, we provide a sketch of the proof. Initially focusing on rank-two matrix approximation, we subsequently extend the analysis to general rank problems. The primary distinction between the rank-two scenario and general rank problems lies in the number of rounds of inductive arguments required.

## 4.1 RANK-TWO MATRIX APPROXIMATION

To start with, we first show that when $\sigma_1(\boldsymbol{X}_0) \leq 1/\sqrt{3\eta}$, the GD sequence will quickly enter the region $\mathcal{S}$ defined in (13), and the sequence will remain in $\mathcal{S}$ afterwards. This proves the first property in Theorem 5. Recall that $t_{\text{init},1} = \min\{t \geq 0 \mid \boldsymbol{X}_t \in \mathcal{S}\}$ and $\boldsymbol{X}_t$ is the GD sequence given by (10).

**Lemma 8** *Suppose* $\eta \leq \frac{1}{12\lambda_1}$ *and* $\sigma_1(\boldsymbol{X}_0) \leq \frac{1}{\sqrt{3\eta}}$. *Then* $\boldsymbol{X}_t \in \mathcal{S}$ *for all* $t \geq t_{\text{init},1}$, *where*

$$t_{\text{init},1} = \mathcal{O}\left(\frac{1}{\eta\lambda_1}\log\frac{1}{6\eta\lambda_1}\right) + \mathcal{O}\left(\frac{1}{\eta\Delta}\log\frac{8\lambda_1}{\Delta}\right).$$

Lemma 8 establishes that $\mathcal{S}$ is an absorbing set of GD, indicating that once the sequence enters this set, it will remain there indefinitely. This characteristic allows us to assume $\boldsymbol{X}_t \in \mathcal{S}$ in subsequent analysis.

### 4.1.1 $\sigma_1^2(\mathbf{u}_{1,t})$ rapidly increases above $\Delta/2$

Our next step is to analyze the first row $\boldsymbol{u}_{1,t}$ of $\boldsymbol{X}_t$. This is in sharp contrast to the results in Section 2.2 and 3.1, where the first $r$ rows of $\boldsymbol{X}_t$ are analyzed together. Although using large initialization makes previous analysis infeasible, it is still manageable to examine only the first row of $\boldsymbol{X}_t$. In Lemma 9, we show that $\sigma_1^2(\boldsymbol{u}_{1,t})$ rapidly increases to be larger or equal to $\Delta/2$, and it remains larger than or equal to this threshold afterwards. This implies that the first signal strength will become larger than or equal to $\Delta/2$ after logarithmic steps. It allows us to employ an SNR argument in the subsequent analysis. Furthermore, this lemma aligns with the initial phase of the GD dynamics as illustrated in Figure 1, providing valuable theoretical insights into the behavior of GD.

**Lemma 9** *Let* $\eta \leq \frac{1}{12\lambda_1}$ *and* $\sigma_1(\boldsymbol{X}_0) \leq 1/\sqrt{3\eta}$. *Assume* $\sigma_1(\boldsymbol{u}_{1,t_{\text{init},1}}) > 0$. *Then*

$$\sigma_1^2(\boldsymbol{u}_{1,t}) \geq \Delta/2, \quad \forall t \geq t_{\text{init},1} + T_{\boldsymbol{u}_1},$$

*where*

$$T_{\boldsymbol{u}_1} = \mathcal{O}\left(\frac{4}{\eta\Delta}\log\frac{\Delta}{2\sigma_1^2(\boldsymbol{u}_{1,t_{\text{init},1}})}\right).$$

### 4.1.2 SNR converges linearly towards infinity and $\sigma_1^2(\mathbf{u}_{1,t})$ converges

Once $\sigma_1^2(\boldsymbol{u}_{1,t})$ exceeds $\Delta/2$, then we can employ an SNR argument similar to (8). Specifically, we pick the SNR as

$$\frac{\sigma_1^2(\boldsymbol{u}_{1,t})}{\sigma_1(\boldsymbol{u}_{1,t}\boldsymbol{K}_{1,t}^\top)},$$

and show that it converges linearly towards infinity, where $\boldsymbol{K}_{1,t}$ is the 2-to-$d$-th rows of $\boldsymbol{X}_t$. Since $\sigma_1^2(\boldsymbol{u}_{1,t})$ is in the interval $[\Delta/2, 2\lambda_1]$ by Lemma 8 and 9, we can show that the noise strength $\sigma_1(\boldsymbol{u}_{1,t}\boldsymbol{K}_{1,t}^\top)$ diminishes to zero fast. In particular, if $\boldsymbol{u}_{1,t}\boldsymbol{K}_{1,t}^\top = \boldsymbol{0}$, then the dynamics of $\boldsymbol{u}_{1,t}$ becomes

$$\boldsymbol{u}_{1,t+1} = \boldsymbol{u}_{1,t} + \eta\lambda_1\boldsymbol{u}_{1,t} - \eta\sigma_1^2(\boldsymbol{u}_{1,t})\boldsymbol{u}_{1,t}.$$

This update rule implies the fast convergence of $\sigma_1^2(\boldsymbol{u}_{1,t})$ to $\lambda_1$. Generally, when the term $\boldsymbol{u}_{1,t}\boldsymbol{K}_{1,t}^\top$ is close to zero, the dynamics of $\boldsymbol{u}_{1,t}$ will mimic the above iteration. Following this, we can establish the fast convergence of $\sigma_1^2(\boldsymbol{u}_{1,t})$ to $\lambda_1$. These results, established in Lemma 10, relate to Property 3 in Theorem 5.

**Lemma 10** *Let* $\eta \leq \frac{\Delta}{100\lambda_1^2}$ *and assume* $\sigma_1(\boldsymbol{X}_0) \leq 1/\sqrt{3\eta}$ *and* $\sigma_1(\boldsymbol{u}_{1,0}) > 0$. *Then for all* $t \geq t_1$, *we have*

$$\sigma_1(\boldsymbol{u}_{1,t}\boldsymbol{K}_{1,t}^\top) \leq (1 - \eta\Delta/6)^{t-t_1}$$

*where* $t_1 = t_{\text{init},1} + T_{\boldsymbol{u}_1} + t^*$, $T_{\boldsymbol{u}_1}$ *is given in Lemma 9, and* $t^*$ *is a constant defined in* (15). *In addition, let*

$$p_{1,t} = \lambda_1 - \sigma_1^2(\boldsymbol{u}_{1,t})$$

*be the error term. Then for all* $t \geq t_1$, *we have*

$$|p_{1,t}| \leq (2\lambda_1 + 24\frac{24r}{\eta\Delta}) \cdot (1 - \eta\Delta/8)^{t-t_1}.$$

The above result implies the rapid convergence of the first signal. This convergence is independent of the behavior of subsequent signals, a phenomenon known as implicit incremental learning. Furthermore, this result corresponds to the second phase of the GD dynamics, as depicted in Figure 1, offering a substantial theoretical explanation.

### 4.1.3 Transition assumption and induction

Lemma 10 shows that the magnitude $\sigma_1(\boldsymbol{u}_{1,t}\boldsymbol{K}_{1,t}^\top)$ diminishes linearly to zero. This motivates us to decouple the original matrix factorization problem into two sub-problems. For

the first sub-problem, we study the convergence of the first row of $X_t$, which has been presented in previous section. In the second sub-problem, we examine $K_{1,t}$, the 2-to-$d$-th rows of $X_t$. Such decoupling is exact when $u_{1,t}K_{1,t}^\top = 0$, and under this condition, the update rule of $K_{1,t}$ becomes

$$K_{1,t} = K_{1,t-1} + \eta(\Gamma_1 - K_{1,t-1}K_{1,t-1}^\top)K_{1,t-1},$$

where $\Gamma_1 = \text{diag}(\lambda_2, \ldots, \lambda_d)$. This is congruent with the GD update rule of $X_t$ as in (10), and hence an inductive argument could be applied.

Generally, when the noise term $\sigma_1(u_{1,t}K_{1,t}^\top)$ only decreases fast but does not reach zero, one should check whether $u_{1,t}K_{1,t}^\top$ is negligible (in the analysis of $u_{2,t}$). Specifically, if $\sigma_1(u_{2,t})$ is not always decreasing at the same speed as $\sigma_1(u_{1,t}K_{1,t}^\top)$, then we can apply the above inductive argument. To formalize this intuition, we introduce a variable $t_1^*$, which is defined as the smallest integer such that

$$r(1 - \eta\Delta/6)^{t_1^*} \leq \sqrt{\frac{\Delta}{8}} \min\{\sigma_1(u_{2,t_1+t_1^*}), \sqrt{\frac{\Delta}{2}}\}, \quad (19)$$

where $t_1$ is defined in Lemma 10. Recall that for all $t \geq t_1$,

$$\sigma_1(u_{1,t}K_{1,t}^\top) \leq (1 - \eta\Delta/6)^{t-t_1}.$$

Thus, (19) essentially compares the second signal strength $\sigma_1(u_{2,\cdot})$ with an upper bound on the noise $\sigma_1(u_{1,t}K_{1,t}^\top)$. When (19) holds, we find that the noise term is negligible, and thus a similar result as Lemma 9 can be established for the second signal $\sigma_1(u_{2,\cdot})$, leading to Lemma 11.

**Lemma 11** *Suppose the conditions of Lemma 10 holds. Let $t_{\text{init},2} = t_1 + t_1^*$, where $t_1$ is given by Lemma 10 and $t_1^*$ is given by* (19). *Suppose $t_1^* < \infty$. Then*

$$\sigma_1^2(u_{2,t}) \geq \Delta/2, \quad \forall t \geq t_{\text{init},2} + T_{u_2},$$

*where*

$$T_{u_2} = \mathcal{O}\left(\frac{4}{\eta\Delta} \log \frac{\Delta}{2\sigma_1^2(u_{2,t_{\text{init},2}})}\right).$$

In Lemma 11, we assume $t_1^* < \infty$, which relates to Assumption 4. If we assume $t_1^* = \mathcal{O}(\log(d))$ as in Assumption 6, then we can show that $T_{u_2} = \mathcal{O}(\log(d))$ as well. While we have not theoretically characterized the quantity $t_1^*$, our theory is still insightful in the following sense.

- First, the term $\sigma_1(u_{1,t}K_{1,t}^\top)$ is shown to decay to zero linearly fast while $\sigma_1^2(u_{2,t})$ does not seem to possess similar theories. Hence, we may expect that the time point $t_1^*$ is not large.
- Second, $t_1^*$ characterizes the time when the GD sequence escapes from the saddle points[7]. This time is inevitable

---

[7]Any stationary point with $u_2 = 0$ is a saddle point. Hence, if the GD sequence $X_t$ converges with $t_1^* = \infty$, then it must converge to a saddle point.

for the GD sequence converging to the global minima. Even we do not provide an upper bound on $t_1^*$, we know the convergence behavior of GD during this time. Notably, during this time, both $\sigma_1(u_{1,t}K_{1,t}^\top)$ and $\sigma_1^2(u_{2,t})$ converge to zero fast.

- Thirdly, during the time $t_1$-$(t_1 + t_1^*)$, while $\sigma_1^2(u_{2,t})$ converges to zero fast, the first signal $\sigma_1^2(u_{1,t})$ still converges to $\lambda_1$, as shown in Lemma 10. This means the convergence of the first signal is not affected by the behaviors of the rest signals, which supports the incremental learning phenomenon – *leading signals first converge even when the rest are stuck by saddle points*.
- Finally, the time $t_1$ to $t_1 + t_1^*$ aligns with the third stage of the GD dynamics as displayed in Figure 1. The experiment shows that the time $t_1^*$ is not too long.

Despite these arguments, there is still a need to examine the duration $t_1^*$ in the future research, which might involve investigating specific initialization mechanisms.

### 4.1.4 Final convergence

Previous analyses indicate that the strengths of both the first and second signals exceed $\Delta/2$, and the corresponding noise components decay geometrically. A simple verification shows that the GD sequence $X_t$ will quickly enter the region $\mathcal{R}$, which is defined in (11). Then by the local linear convergence of GD in Theorem 2, we shall complete the characterization of the GD sequence's convergence to the global minima. This final stage aligns with the fourth stage of the GD dynamics as illustrated in Figure 1.

### 4.2 GENERAL RANK MATRIX APPROXIMATION

It is straightforward to extend the rank-two case to the general rank case. The key point is to repeat the inductive arguments for $(r - 1)$ rather than one times.

Similar to the rank-two case, we will now successively show that $\sigma_1^2(u_{k,t})$ surpasses $\Delta/2$ and $\sigma_1(u_{k,t}K_{k,t}^\top)$ diminishes linearly to zero for all $k \leq r$. Moreover, we will show that $\sigma_1^2(u_{k,t})$ converges to $\lambda_k$ after certain iterations. Once the first $r$ rows of $X_t$ are all analyzed, we can show that the sequence $X_t$ quickly enters the region $\mathcal{R}$ defined in (11). By invoking the local linear convergence theorem, we will conclude the proof.

Our analysis consistently uses the SNR argument, although the specific SNR definitions vary.

- For analyzing the $k$-th signal strength in Theorem 5, we examine the SNR given by

$$\frac{\sigma_1^2(u_{k,t})}{\sigma_1(u_{k,t}K_{k,t}^\top)}.$$

We will prove both the diminishing of $\sigma_1(\boldsymbol{u}_{k,t}\boldsymbol{K}_{k,t}^\top)$ to zero and the convergence of $\sigma_1^2(\boldsymbol{u}_{k,t})$ to $\lambda_k$.

- For analyzing the local linear convergence in Theorem 2, we utilize the SNR defined as

$$\sigma_r^2(\boldsymbol{U}_t)/\sigma_1^2(\boldsymbol{J}_t),$$

where $\boldsymbol{U}, \boldsymbol{J}$ are defined in Section 2.2. We will prove the linear convergence of $\boldsymbol{J}$ to zero, as well as the fast convergence of $\boldsymbol{U}_t$ to the target matrix.

## 5 CONCLUDING REMARKS

This paper presents a comprehensive analysis of the trajectory of GD for matrix factorization problems, with a partcular focus on large initialization. By employing both an SNR argument and inductive reasoning, we deepen the investigation and uncover that even with large initialization, GD still exhibit an incremental learning phenomenon. We anticipate that these insights will stimulate further research in related domains.

This study presents several limitations that naturally suggest avenues for future research.

- First, we have not established an upper bound for $t_k^*$ defined in (16). Determining an effective upper bound is crucial, and exploring potential negative results in this context could also be insightful.
- Second, our analysis assumes strictly decreasing top eigenvalues. Extending the findings to matrices with possibly equal eigenvalues requires additional research.
- Third, our analysis is confined to the simplest matrix factorization scenario. Exploring these results in more complex settings, such as matrix sensing where $\boldsymbol{\Sigma}$ is accessible only through linear measurements, would be particularly compelling. Given that our dynamic analysis is sensitive to noise, this generalization may be challenging.
- Last, investigating GD in deep matrix factorization also presents a significant research opportunity. It remains unclear how large initialization impacts the GD trajectory in such a complex case.

## ACKNOWLEDGEMENT

Hengchao Chen and Qiang Sun are supported in part by an NSERC Dicovery Grant (RGPIN-2018-06484), a Data Sciences Institute Catalyst Grant, and a computing grant from Compute Canada.

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

# Gradient descent in matrix factorization: Understanding large initialization (Supplementary Material)

**Hengchao Chen**[1]       **Xin Chen**[2]       **Mohamad Elmasri**[1]       **Qiang Sun**[1,3]

[1]University of Toronto
[2]Princeton University
[3]MBZUAI

# APPENDIX

## A   PRELIMINARY

In this section, we present preliminary to our studied formulation.

### A.1   MATRIX SENSING AND IT POPULATION VERSION

In this section, we review matrix sensing problems [Recht et al., 2010, Li et al., 2018] and derive its population version as a matrix factorization problem [Chi et al., 2019]. In matrix sensing problems, we aim to recover an unknown symmetric positive semidefinite matrix $\boldsymbol{\Sigma} \in \mathbb{R}^{d \times d}$ from a set of linear measurements

$$y_i = \langle \boldsymbol{\Sigma}, \boldsymbol{A}_i \rangle, \quad i = 1, \ldots, m.$$

Here $\{\boldsymbol{A}_i\} \subseteq \mathbb{R}^{d \times d}$ are sensing matrices known *a priori*. A standard assumption is to require that $\boldsymbol{\Sigma}$ is of rank $r \ll d$ and $\boldsymbol{A}_i$ has independent $\mathcal{N}(0, 1)$ entries. Under such assumptions, one common strategy is to employ matrix factorization, and solve the following least square minimizing problem:

$$\hat{\boldsymbol{\Sigma}} = \hat{\boldsymbol{X}} \hat{\boldsymbol{X}}^\top, \quad \hat{\boldsymbol{X}} = \underset{\boldsymbol{X} \in \mathbb{R}^{d \times r}}{\arg \min} f(\boldsymbol{X}) \overset{\text{def}}{=} \frac{1}{4m} \sum_{i=1}^m (y_i - \langle \boldsymbol{A}_i, \boldsymbol{X} \boldsymbol{X}^\top \rangle)^2.$$

Starting from an initial point $\boldsymbol{X}_0 \in \mathbb{R}^{d \times r}$, the gradient descent algorithm will update $\boldsymbol{X}_t$ as follows

$$\boldsymbol{X}_t = \boldsymbol{X}_{t-1} - \eta \nabla f(\boldsymbol{X}_{t-1}),$$

where $\nabla f(\boldsymbol{X})$ is the gradient of $f$ given by

$$\nabla f(\boldsymbol{X}) = -\frac{1}{m} \sum_{i=1}^m (y_i - \langle \boldsymbol{A}_i, \boldsymbol{X} \boldsymbol{X}^\top \rangle) \boldsymbol{A}_i \boldsymbol{X} = -\frac{1}{m} \sum_{i=1}^m \langle \boldsymbol{A}_i, \boldsymbol{\Sigma} - \boldsymbol{X} \boldsymbol{X}^\top \rangle \cdot \boldsymbol{A}_i \boldsymbol{X}.$$

Using statistical concentration analysis [Recht et al., 2010], we know that the gradient $\nabla f(\boldsymbol{X})$ is approximately

$$\nabla f(\boldsymbol{X}) \approx \mathbb{E} \nabla f(\boldsymbol{X}) = -(\boldsymbol{\Sigma} - \boldsymbol{X} \boldsymbol{X}^\top) \boldsymbol{X}.$$

Thus, the matrix sensing is related to the following population version:

$$\boldsymbol{X}_t = \boldsymbol{X}_{t-1} + \eta (\boldsymbol{\Sigma} - \boldsymbol{X}_t \boldsymbol{X}_t^\top) \boldsymbol{X}_t,$$

which is the GD update rule (10) for matrix factorization problems. Many works [Recht et al., 2010, Li et al., 2018, Zhu et al., 2018, 2021] build on the intuition that for sufficiently large $m$, matrix sensing is approximately the matrix factorization.

These works rely on the restricted isotropy property (RIP) established in Recht et al. [2010]. We can use this property and our analysis in Section 2.2 to derive the local linear convergence in the matrix sensing problem. However, for the case of large initialization, RIP seems not sufficient. Therefore, extending matrix factorization to matrix sensing requires further statistical analysis.

## A.2 INVARIANCE TO ORTHOGONAL ROTATIONS

In this section, we elaborate why there is no loss of generality to assume that $\boldsymbol{\Sigma}$ is a diagonal matrix. Specifically, when $\boldsymbol{\Sigma}$ is a general symmetric positive semidefinite matrix, we can write $\boldsymbol{\Sigma} = \boldsymbol{U}\boldsymbol{\Lambda}\boldsymbol{U}^\top$ by the eigen-decomposition, where $\boldsymbol{U} \in \mathbb{R}^{d \times d}$ is an orthogonal matrix and $\boldsymbol{\Lambda}$ is a diagonal matrix. Suppose the GD sequence is given by

$$\boldsymbol{X}_t = \boldsymbol{X}_{t-1} + \eta(\boldsymbol{\Sigma} - \boldsymbol{X}_{t-1}\boldsymbol{X}_{t-1}^\top)\boldsymbol{X}_{t-1}.$$

Then we consider the transformation $\boldsymbol{Y}_t = \boldsymbol{U}\boldsymbol{X}_t$, which leads to

$$\boldsymbol{Y}_t = \boldsymbol{Y}_{t-1} + \eta(\boldsymbol{\Lambda} - \boldsymbol{Y}_{t-1}\boldsymbol{Y}_{t-1}^\top)\boldsymbol{Y}_{t-1}.$$

This reduces to the case where $\boldsymbol{\Lambda}$ is diagonal. In addition, the convergences of $\boldsymbol{Y}_t$ and $\boldsymbol{X}_t$ are associated, and the random initialization of $\boldsymbol{Y}_0$ and $\boldsymbol{X}_0$ are associated up to an orthogonal matrix. Consequently, there is no loss of generality to assume that $\boldsymbol{\Sigma}$ is diagonal.

# B PROOF OF THEOREM 2

Our proof of Theorem 2 consists of three steps.

- First, we show that $\mathcal{R}$ is an absorbing region for GD. Here a set is regarded as an absorbing set if the GD sequence remains within the set after its first entrance.
- Next, we show that $\sigma_1(\boldsymbol{J}_t)$ converges to zero at a linear rate, employing an SNR argument.
- Finally, we establish the linear convergence to the global minima.

Before diving deeper, we first write down the update rules for $\boldsymbol{U}_t$ and $\boldsymbol{J}_t$. By (10), we have

$$\boldsymbol{U}_{t+1} = \boldsymbol{U}_t + \eta\boldsymbol{\Lambda}_r\boldsymbol{U}_t - \eta\boldsymbol{U}_t\boldsymbol{X}_t^\top\boldsymbol{X}_t, \tag{20}$$

$$\boldsymbol{J}_{t+1} = \boldsymbol{J}_t + \eta\boldsymbol{\Lambda}_{\text{res}}\boldsymbol{J}_t - \eta\boldsymbol{J}_t\boldsymbol{X}_t^\top\boldsymbol{X}_t, \tag{21}$$

where $\boldsymbol{\Lambda}_r = \text{diag}(\lambda_1, \ldots, \lambda_r)$ and $\boldsymbol{\Lambda}_{\text{res}} = \text{diag}(\lambda_{r+1}, \ldots, \lambda_d)$. Note that $\boldsymbol{\Sigma}_r = \text{diag}(\boldsymbol{\Lambda}_r, \boldsymbol{0})$.

## B.1 THE GD SEQUENCE REMAINS IN $\mathcal{R}$

Lemma 12 shows that $\mathcal{R}$ is an absorbing region for GD.

**Lemma 12** *Suppose $\eta \leq \frac{\Delta^2}{36\lambda_1^3}$ and $\boldsymbol{X}_t \in \mathcal{R}$. Then $\boldsymbol{X}_{t'} \in \mathcal{R}$ for all $t' \geq t$.*

**Proof** This lemma is proved by induction. Suppose $\boldsymbol{X}_t \in \mathcal{R}$.

- By Lemma 13 and $\sigma_1^2(\boldsymbol{X}_t) \leq 2\lambda_1$, we get $\sigma_1^2(\boldsymbol{X}_{t+1}) \leq 2\lambda_1$.
- By Lemma 14, $\sigma_1^2(\boldsymbol{X}_t) \leq 2\lambda_1$, and $\sigma_1^2(\boldsymbol{J}_t) \leq \lambda_r - \Delta/2$, we get $\sigma_1^2(\boldsymbol{J}_{t+1}) \leq \lambda_r - \Delta/2$.
- By Lemma 15 and $\boldsymbol{X}_t \in \mathcal{R}$, we get $\sigma_r^2(\boldsymbol{U}_{t+1}) \geq \Delta/4$ and thus $\boldsymbol{X}_{t+1} \in \mathcal{R}$.

By induction, we conclude that $\boldsymbol{X}_{t'} \in \mathcal{R}$ for all $t' \geq t$. ∎

### B.1.1 Technical lemmas

In this section, we summarize technical lemmas used in the proof of Lemma 12.

Lemma 13 delineates the first category of absorbing sets for GD, denoted as

$$\mathcal{S}_1 = \{ \boldsymbol{X} \in \mathbb{R}^{d \times r} \mid \sigma_1(\boldsymbol{X}) \leq a \},$$

valid for any $a \in [\sqrt{\lambda_1}, 1/\sqrt{3\eta}]$.

**Lemma 13** *Suppose $\eta \leq \frac{1}{3\lambda_1}$ and $a \in [\sqrt{\lambda_1}, 1/\sqrt{3\eta}]$. If $\sigma_1(\boldsymbol{X}_t) \leq a$, then $\sigma_1(\boldsymbol{X}_{t'}) \leq a, \forall t' \geq t$.*

**Proof** Lemma 16 states that if $\sigma_1(\boldsymbol{X}_t) \leq 1/\sqrt{3\eta}$, then the following inequality holds

$$\sigma_1(\boldsymbol{X}_{t+1}) \leq (1 + \eta\lambda_1 - \eta\sigma_1^2(\boldsymbol{X}_t)) \cdot \sigma_1(\boldsymbol{X}_t).$$

- If $\sqrt{\lambda_1} \leq \sigma_1(\boldsymbol{X}_t) \leq a$, the above inequality implies that $\sigma_1(\boldsymbol{X}_{t+1}) \leq \sigma_1(\boldsymbol{X}_t) \leq a$.
- If $\sigma_1(\boldsymbol{X}_t) \leq \sqrt{\lambda_1} \leq a$, it follows that

$$\sigma_1(\boldsymbol{X}_{t+1}) \leq (1 + \eta\lambda_1 - \eta\lambda_1)\sqrt{\lambda_1} \leq a.$$

  This uses the fact that $g_1(s) = (1 + \eta\lambda_1 - \eta s^2)s$ is increasing on $[0, 1/\sqrt{3\eta}]$.

By induction, we have $\sigma_1(\boldsymbol{X}_{t'}) \leq a$ for all $t' \geq t$. ∎

Lemma 14 demonstrates that if $\sigma_1(\boldsymbol{X}_t) \leq \sqrt{2\lambda_1}$, $\sigma_1^2(\boldsymbol{J}_t) \leq a$, and $a \geq \lambda_{r+1}$, then $\sigma_1^2(\boldsymbol{J}_{t+1}) \leq a$. Combining with Lemma 13, it implies that

$$\mathcal{S}_2 = \{ \boldsymbol{X} = \begin{pmatrix} \boldsymbol{U} \\ \boldsymbol{J} \end{pmatrix} \in \mathbb{R}^{d \times r} \mid \sigma_1(\boldsymbol{X}) \leq \sqrt{2\lambda_1}, \sigma_1^2(\boldsymbol{J}) \leq a \}$$

is an absorbing set for GD, provided that $a \geq \lambda_{r+1}$ and $\eta \leq \frac{1}{12\lambda_1}$. Here $\boldsymbol{U}$ and $\boldsymbol{J}$ are the top $r$ rows and the $(r+1)$-to-$d$-th rows of $\boldsymbol{X}$ respectively.

**Lemma 14** *Suppose $\eta \leq \frac{1}{12\lambda_1}$, $\sigma_1^2(\boldsymbol{X}_t) \leq 2\lambda_1$, and $a \geq \lambda_{r+1}$. If $\sigma_1^2(\boldsymbol{J}_t) \leq a$, then $\sigma_1^2(\boldsymbol{J}_{t+1}) \leq a$.*

**Proof** By Lemma 17, we have

$$\sigma_1(\boldsymbol{J}_{t+1}) \leq (1 + \eta(\lambda_{r+1} - \sigma_1^2(\boldsymbol{J}_t))) \cdot \sigma_1(\boldsymbol{J}_t).$$

- If $\lambda_{r+1} < \sigma_1^2(\boldsymbol{J}_t) \leq a$, then it follows that $\sigma_1^2(\boldsymbol{J}_{t+1}) \leq \sigma_1^2(\boldsymbol{J}_t) \leq a$.
- If $\sigma_1^2(\boldsymbol{J}_t) \leq \lambda_{r+1} \leq a$, then

$$\sigma_1^2(\boldsymbol{J}_{t+1}) \leq (1 + \eta(\lambda_{r+1} - \lambda_{r+1}))^2 \lambda_{r+1} \leq a.$$

  This uses the observation that $g_2(s) = (1 + \eta(\lambda_{r+1} - s^2))s$ is increasing on $[0, 1/\sqrt{3\eta}]$.

This concludes the proof. ∎

Lemma 15 is the last piece needed to show that region $\mathcal{R}$ is an absorbing set for GD.

**Lemma 15** *Suppose $\eta \leq \frac{\Delta^2}{32\lambda_1^3}$, $\sigma_1(\boldsymbol{X}_t) \leq \sqrt{2\lambda_1}$, and $\sigma_1^2(\boldsymbol{J}_t) \leq \lambda_r - \Delta/2$. If $\sigma_r^2(\boldsymbol{U}_t) \geq \Delta/4$, then $\sigma_r^2(\boldsymbol{U}_{t+1}) \geq \Delta/4$.*

**Proof** Since $\eta \leq \frac{1}{32\lambda_1}$ and $\sigma_1^2(\boldsymbol{J}_t) \leq \lambda_r - \Delta/2$, by Lemma 18, we have

$$\sigma_r^2(\boldsymbol{U}_{t+1}) \geq (1 + \eta\Delta - 2\eta\sigma_r^2(\boldsymbol{U}_t)) \cdot \sigma_r^2(\boldsymbol{U}_t) - 4\eta^2\lambda_1^3.$$

Since $g_3(s) = (1 + \eta\Delta - 2\eta s)s$ is increasing on $(-\infty, \frac{1}{4\eta}]$ and $\frac{\Delta}{4} \leq \sigma_r^2(U_t) \leq 2\lambda_1 \leq \frac{1}{4\eta}$, we have

$$\sigma_r^2(U_{t+1}) \geq (1 + \frac{\eta\Delta}{2}) \cdot \frac{\Delta}{4} - 4\eta^2\lambda_1^3 \geq \frac{\Delta}{4},$$

where the last inequality uses $\eta \leq \frac{\Delta^2}{32\lambda_1^3}$. ∎

The following lemmas give certain singular value analysis that are used in prior lemmas and subsequent analysis. Lemma 16 establishes an upper bound for $\sigma_1(X_{t+1})$.

**Lemma 16** *If $\sigma_1(X_t) \leq 1/\sqrt{3\eta}$, then we have*

$$\sigma_1(X_{t+1}) \leq (1 + \eta\lambda_1 - \eta\sigma_1^2(X_t)) \cdot \sigma_1(X_t).$$

**Proof** By the singular value inequality and (10),

$$\begin{aligned}
\sigma_1(X_{t+1}) &\leq \sigma_1(X_t(I_r - \eta X_t^\top X_t)) + \eta\sigma_1(\Sigma X_t) \\
&\leq \sigma_1(X_t(I_r - \eta X_t^\top X_t)) + \eta\lambda_1\sigma_1(X_t),
\end{aligned} \tag{22}$$

where we use $\sigma_1(\Sigma) = \lambda_1$. Observe that all $r$ singular values of $X_t(I_r - \eta X_t^\top X_t)$ are given by

$$(1 - \eta\sigma_i^2(X_t)) \cdot \sigma_i(X_t), \quad i = 1, \dots, r,$$

since $\eta\sigma_1^2(X_t) \leq 1$. The function $g_4(s) = (1 - \eta s^2)s$ is increasing on $[0, 1/\sqrt{3\eta}]$. Hence, the fact $0 \leq \sigma_i(X_t) \leq \sigma_1(X_t) \leq 1/\sqrt{3\eta}$ implies that

$$\sigma_1(X_t(I_r - \eta X_t^\top X_t)) = (1 - \eta\sigma_1^2(X_t)) \cdot \sigma_1(X_t).$$

Substituting this equality into (22), we conclude the proof. ∎

Lemma 17 gives an upper bound for $\sigma_1(J_{t+1})$.

**Lemma 17** *Suppose $\eta \leq \frac{1}{12\lambda_1}$ and $\sigma_1(X_t) \leq \sqrt{2\lambda_1}$, then we have*

$$\sigma_1(J_{t+1}) \leq (1 + \eta(\lambda_{r+1} - \sigma_1^2(J_t) - \sigma_r^2(U_t))) \cdot \sigma_1(J_t).$$

**Proof** The update rule (21) of $J_{t+1}$ can be decomposed as follows:

$$J_{t+1} = \underbrace{\frac{1}{2}J_t - \eta J_t J_t^\top J_t}_{B} + \underbrace{(\frac{1}{4}I_{d-r} + \eta\Lambda_{\text{res}})J_t}_{C} + \underbrace{J_t(\frac{1}{4}I_r - \eta U_t^\top U_t)}_{D}.$$

By the singular value inequality,

$$\sigma_1(J_{t+1}) \leq \sigma_1(B) + \sigma_1(C) + \sigma_1(D).$$

Observe that all singular values of $B$ are given by

$$\sigma_i(J_t)/2 - \eta\sigma_i^3(J_t), \quad i = 1, \dots, d - r.$$

Since $g_5(s) = s/2 - \eta s^3$ is increasing on $[0, 1/\sqrt{6\eta}]$, the condition $\sigma_i(J_t) \leq \sigma_1(J_t) \leq \sqrt{2\lambda_1} \leq 1/\sqrt{6\eta}$ implies that

$$\sigma_1(B) = \sigma_1(J_t)/2 - \eta\sigma_1^3(J_t).$$

For the second term $C$, it follows from the singular value inequality that

$$\sigma_1(C) \leq \sigma_1(\frac{1}{4}I_{d-r} + \eta\Lambda_{\text{res}})\sigma_1(J_t) \leq (1/4 + \eta\lambda_{r+1})\sigma_1(J_t),$$

where the second inequality uses $\eta\sigma_1(\Lambda_{\text{res}}) \leq \eta\lambda_1 \leq 1/4$. For the third term $D$, since $\eta\sigma_1^2(U_t) \leq 2\eta\lambda_1 \leq 1/4$, we have

$$\sigma_1(D) \leq (1/4 - \eta\sigma_r^2(U_t))\sigma_1(J_t).$$

Finally, we conclude the proof by combining the analysis of $B, C$, and $D$. ∎

Lemma 18 provides a lower bound for $\sigma_r^2(U_{t+1})$.

**Lemma 18** *Suppose $\eta \leq \frac{1}{32\lambda_1}$ and $\sigma_1(\boldsymbol{X}_t) \leq \sqrt{2\lambda_1}$, then we have*

$$\sigma_r^2(\boldsymbol{U}_{t+1}) \geq (1 + 2\eta(\lambda_r - \sigma_1^2(\boldsymbol{J}_t) - \sigma_r^2(\boldsymbol{U}_t))) \cdot \sigma_r^2(\boldsymbol{U}_t) - 4\eta^2\lambda_1^3.$$

**Proof** Substituting the update rule (20) of $\boldsymbol{U}_{t+1}$ into $\boldsymbol{U}_{t+1}\boldsymbol{U}_{t+1}^\top$, we get

$$\begin{aligned}
\boldsymbol{U}_{t+1}\boldsymbol{U}_{t+1}^\top &= (\boldsymbol{U}_t - \eta\boldsymbol{U}_t\boldsymbol{X}_t^\top\boldsymbol{X}_t + \eta\boldsymbol{\Lambda}_r\boldsymbol{U}_t) \cdot (\boldsymbol{U}_t - \eta\boldsymbol{U}_t\boldsymbol{X}_t^\top\boldsymbol{X}_t + \eta\boldsymbol{\Lambda}_r\boldsymbol{U}_t)^\top \\
&= \boldsymbol{B} + \boldsymbol{C} - \eta^2\boldsymbol{R}_1 + \eta^2\boldsymbol{R}
\end{aligned}$$

where

$$\begin{aligned}
\boldsymbol{B} &= \boldsymbol{U}_t(\frac{1}{2}\boldsymbol{I}_r - 2\eta\boldsymbol{X}_t^\top\boldsymbol{X}_t)\boldsymbol{U}_t^\top, \\
\boldsymbol{C} &= (\frac{1}{\sqrt{2}}\boldsymbol{I}_r + \sqrt{2}\eta\boldsymbol{\Lambda}_r)\boldsymbol{U}_t\boldsymbol{U}_t^\top(\frac{1}{\sqrt{2}}\boldsymbol{I}_r + \sqrt{2}\eta\boldsymbol{\Lambda}_r), \\
\boldsymbol{R}_1 &= 2\boldsymbol{\Lambda}_r\boldsymbol{U}_t\boldsymbol{U}_t^\top\boldsymbol{\Lambda}_r, \\
\boldsymbol{R} &= (\boldsymbol{\Lambda}_r\boldsymbol{U}_t - \boldsymbol{U}_t\boldsymbol{X}_t^\top\boldsymbol{X}_t)(\boldsymbol{\Lambda}_r\boldsymbol{U}_t - \boldsymbol{U}_t\boldsymbol{X}_t^\top\boldsymbol{X}_t)^\top.
\end{aligned}$$

Here $\boldsymbol{B}$ is positive semi-definite (PSD) since $2\eta\sigma_1^2(\boldsymbol{X}_t) \leq 4\eta\lambda_1 \leq 1/2$ and $\boldsymbol{C}, \boldsymbol{R}_1, \boldsymbol{R}$ are all PSD. By the eigenvalue inequality and the equivalence between eigenvalues and singular values of a PSD matrix, we have

$$\begin{aligned}
\sigma_r^2(\boldsymbol{U}_{t+1}) &\geq \sigma_r(\boldsymbol{B}) + \sigma_r(\boldsymbol{C}) - \eta^2\sigma_1(\boldsymbol{R}_1) + \eta^2\sigma_r(\boldsymbol{R}) \\
&\geq \sigma_r(\boldsymbol{B}) + \sigma_r(\boldsymbol{C}) - \eta^2\sigma_1(\boldsymbol{R}_1).
\end{aligned} \tag{23}$$

For the first term $\boldsymbol{B}$, we decompose it into two terms:

$$\boldsymbol{B} = \underbrace{\boldsymbol{U}_t((\frac{1}{2} - 2\eta\sigma_1^2(\boldsymbol{J}_t)) \cdot \boldsymbol{I}_r - 2\eta\boldsymbol{U}_t^\top\boldsymbol{U}_t)\boldsymbol{U}_t^\top}_{\boldsymbol{B}_1} + 2\eta \cdot \underbrace{\boldsymbol{U}_t(\sigma_1^2(\boldsymbol{J}_t) \cdot \boldsymbol{I}_r - \boldsymbol{J}_t^\top\boldsymbol{J}_t)\boldsymbol{U}_t^\top}_{\boldsymbol{B}_2}.$$

The inequality $2\eta(\sigma_1^2(\boldsymbol{J}_t) + \sigma_1^2(\boldsymbol{U}_t)) \leq 8\eta\lambda_1 \leq 1/2$ implies that $\boldsymbol{B}_1$ is PSD. Since $\boldsymbol{B}_2$ is also PSD, we have $\sigma_r(\boldsymbol{B}) \geq \sigma_r(\boldsymbol{B}_1)$. To determine $\sigma_r(\boldsymbol{B}_1)$, we write the singular values of $\boldsymbol{B}_1$ as

$$(\frac{1}{2} - 2\eta\sigma_1^2(\boldsymbol{J}_t)) \cdot \sigma_i^2(\boldsymbol{U}_t) - 2\eta\sigma_i^4(\boldsymbol{U}_t), \; i = 1, \ldots, r.$$

Since $1/2 - 2\eta\sigma_1^2(\boldsymbol{J}_t) \geq 1/4$, the function $g_6(s) = (1/2 - 2\eta\sigma_1^2(\boldsymbol{J}_t))s - 2\eta s^2$ is increasing on $(-\infty, \frac{1}{16\eta}]$. Then the inequality $\sigma_i^2(\boldsymbol{U}_t) \leq \sigma_1^2(\boldsymbol{U}_t) \leq 2\lambda_1 \leq \frac{1}{16\eta}$ implies that

$$\sigma_r(\boldsymbol{B}_1) = (\frac{1}{2} - 2\eta(\sigma_1^2(\boldsymbol{J}_t) + \sigma_r^2(\boldsymbol{U}_t))) \cdot \sigma_r^2(\boldsymbol{U}_t).$$

For the second term $\boldsymbol{C}$, we have

$$\sigma_r(\boldsymbol{C}) \geq \sigma_r^2(\frac{1}{\sqrt{2}}\boldsymbol{I}_r + \sqrt{2}\eta\boldsymbol{\Lambda}_r)\sigma_r^2(\boldsymbol{U}_t) \geq (\frac{1}{2} + 2\eta\lambda_r)\sigma_r^2(\boldsymbol{U}_t).$$

For the third term $\boldsymbol{R}_1$, since $\sigma_1^2(\boldsymbol{X}_t) \leq 2\lambda_1$, we have

$$\sigma_1(\boldsymbol{R}_1) \leq 4\lambda_1^3.$$

Finally, substituting the analysis of $\boldsymbol{B}, \boldsymbol{C}, \boldsymbol{R}_1$ into (23) gives the desired result. ∎

## B.2 $\sigma_1(\boldsymbol{J}_t)$ CONVERGES TO ZERO LINEARLY VIA AN SNR ARGUMENT

Lemma 19 shows that if $\boldsymbol{X}_0 \in \mathcal{R}$, then $\sigma_1(\boldsymbol{J}_t)$ will diminish to zero at a geometric rate. A key step of the analysis is to examine the SNR $\frac{\sigma_r^2(\boldsymbol{U}_t)}{\sigma_1^2(\boldsymbol{J}_t)}$. Our analysis extends the rank-one case in Section 2.1 to a general rank scenario.

**Lemma 19** *Suppose $\eta \leq \Delta^2/(32\lambda_1^3)$ and $\boldsymbol{X}_0 \in \mathcal{R}$. Then, for all $t \geq 0$, we have*

$$\frac{\sigma_1^2(\boldsymbol{J}_{t+1})}{\sigma_r^2(\boldsymbol{U}_{t+1})} \leq (1 - \eta\Delta/3) \cdot \frac{\sigma_1^2(\boldsymbol{J}_t)}{\sigma_r^2(\boldsymbol{U}_t)}.$$

*Hence, $\sigma_1^2(\boldsymbol{J}_t) \leq 8\lambda_1^2(1 - \eta\Delta/3)^t/\Delta$ for all $t$ and $\sigma_1^2(\boldsymbol{J}_t) < \epsilon$ after*

$$T_{\boldsymbol{J}}^\epsilon = \mathcal{O}\left(\frac{3}{\eta\Delta} \log \frac{8\lambda_1^2}{\epsilon\Delta}\right) \quad \textit{iterations.}$$

**Proof** By Lemma 12, we have $\boldsymbol{X}_t \in \mathcal{R}$ for all $t \geq 0$. Then by Lemma 17,

$$\sigma_1^2(\boldsymbol{J}_{t+1}) \leq (1 + 2\eta(\lambda_{r+1} - \sigma_1^2(\boldsymbol{J}_t) - \sigma_r^2(\boldsymbol{U}_t)) + 16\eta^2\lambda_1^2) \cdot \sigma_1^2(\boldsymbol{J}_t)$$
$$\leq (1 - \eta\Delta/2 + 2\eta(\lambda_r - \Delta/2 - \sigma_1^2(\boldsymbol{J}_t) - \sigma_r^2(\boldsymbol{U}_t))) \cdot \sigma_1^2(\boldsymbol{J}_t),$$

where the second inequality follows from $\eta \leq \frac{\Delta}{32\lambda_1^2}$. By Lemma 18,

$$\sigma_r^2(\boldsymbol{U}_{t+1}) \geq (1 + \eta\Delta + 2\eta(\lambda_r - \Delta/2 - \sigma_1^2(\boldsymbol{J}_t) - \sigma_r^2(\boldsymbol{U}_t))) \cdot \sigma_r^2(\boldsymbol{U}_t) - 4\eta^2\lambda_1^3$$
$$\geq (1 + \eta\Delta/2 + 2\eta(\lambda_r - \Delta/2 - \sigma_1^2(\boldsymbol{J}_t) - \sigma_r^2(\boldsymbol{U}_t))) \cdot \sigma_r^2(\boldsymbol{U}_t),$$

where we use $\sigma_r^2(\boldsymbol{U}_t) \geq \Delta/4$ and $\eta \leq \frac{\Delta^2}{32\lambda_1^3}$ in the second inequality. A combination of the above two inequalities gives that

$$\frac{\sigma_1^2(\boldsymbol{J}_{t+1})}{\sigma_r^2(\boldsymbol{U}_{t+1})} \leq \frac{1 - \eta\Delta/2 + 2\eta(\lambda_r - \Delta/2 - \sigma_1^2(\boldsymbol{J}_t) - \sigma_r^2(\boldsymbol{U}_t))}{1 + \eta\Delta/2 + 2\eta(\lambda_r - \Delta/2 - \sigma_1^2(\boldsymbol{J}_t) - \sigma_r^2(\boldsymbol{U}_t))} \cdot \frac{\sigma_1^2(\boldsymbol{J}_t)}{\sigma_r^2(\boldsymbol{U}_t)}.$$

Since the function $g_7(s) = \frac{1 - \eta\Delta/2 + s}{1 + \eta\Delta/2 + s}$ is increasing on $[-1/2, 1/2]$, the condition $-1/2 \leq 2\eta(\lambda_r - \Delta/2 - \sigma_1^2(\boldsymbol{J}_t) - \sigma_r^2(\boldsymbol{U}_t)) \leq 1/2$ implies that

$$\frac{\sigma_1^2(\boldsymbol{J}_{t+1})}{\sigma_r^2(\boldsymbol{U}_{t+1})} \leq \frac{3/2 - \eta\Delta/2}{3/2 + \eta\Delta/2} \cdot \frac{\sigma_1^2(\boldsymbol{J}_t)}{\sigma_r^2(\boldsymbol{U}_t)} \leq (1 - \eta\Delta/3) \cdot \frac{\sigma_1^2(\boldsymbol{J}_t)}{\sigma_r^2(\boldsymbol{U}_t)}.$$

By deduction, we have

$$\sigma_1^2(\boldsymbol{J}_t) \leq (1 - \eta\Delta/3)^t \cdot \sigma_r^2(\boldsymbol{U}_t) \frac{\sigma_1^2(\boldsymbol{J}_0)}{\sigma_r^2(\boldsymbol{U}_0)} \leq (1 - \eta\Delta/3)^t \cdot \frac{8\lambda_1^2}{\Delta},$$

where the second inequality follows from $\sigma_r^2(\boldsymbol{U}_t) \leq 2\lambda_1$, $\sigma_1^2(\boldsymbol{J}_0) \leq \lambda_1$, and $\sigma_r^2(\boldsymbol{U}_0) \geq \Delta/4$. Therefore, for any $\epsilon > 0$, it takes at most $T_{\boldsymbol{J}}^\epsilon = \mathcal{O}(\frac{3}{\eta\Delta} \log \frac{8\lambda_1^2}{\epsilon\Delta})$ iterations to have $\sigma_1^2(\boldsymbol{J}_t) \leq \epsilon$. ∎

## B.3   FINAL CONVERGENCE

For the convergence of $\boldsymbol{X}_t\boldsymbol{X}_t^\top$ to $\boldsymbol{\Sigma}_r$, It remains to show that $\boldsymbol{U}_t\boldsymbol{U}_t^\top$ converges to $\boldsymbol{\Lambda}_r$ fast, where $\boldsymbol{\Lambda}_r = \text{diag}(\lambda_1, \ldots, \lambda_r)$. Equivalently, it suffices to show that $\sigma_1(\boldsymbol{P}_t)$ converges to zero linearly, where $\boldsymbol{P}_t = \boldsymbol{\Lambda}_t - \boldsymbol{U}_t\boldsymbol{U}_t^\top$. This is established in Lemma 20.

**Lemma 20** *Suppose $\eta \leq \Delta^2/(36\lambda_1^3)$ and $\boldsymbol{X}_0 \in \mathcal{R}$. Then, for all $t \geq 0$, we have*

$$\sigma_1(\boldsymbol{P}_{t+1}) \leq \frac{100\lambda_1^2}{\eta\Delta^2}(1 - \eta\Delta/4)^{t+1}.$$

*Hence, for any $\epsilon > 0$, it takes $T_{\boldsymbol{P}}^\epsilon = \mathcal{O}\left(\frac{4}{\eta\Delta} \log \frac{100\lambda_1^2}{\eta\Delta^2\epsilon}\right)$ iterations to reach $\sigma_1(\boldsymbol{P}_t) \leq \epsilon$.*

**Proof** By Lemma 12, $\boldsymbol{X}_t \in \mathcal{R}$ for all $t \geq 0$. Using the notation of $\boldsymbol{P}_t$, (20) can be rewritten as
$$\boldsymbol{U}_{t+1} = \boldsymbol{U}_t + \eta \boldsymbol{P}_t \boldsymbol{U}_t - \eta \boldsymbol{U}_t \boldsymbol{J}_t^\top \boldsymbol{J}_t.$$

By direct calculation, we have
$$\boldsymbol{P}_{t+1} = (\boldsymbol{I}_r - \eta \boldsymbol{U}_t \boldsymbol{U}_t^\top) \boldsymbol{P}_t (\boldsymbol{I}_r - \eta \boldsymbol{U}_t \boldsymbol{U}_t^\top) - \eta^2 (\boldsymbol{P}_t \boldsymbol{U}_t \boldsymbol{U}_t^\top \boldsymbol{P}_t + \boldsymbol{U}_t \boldsymbol{U}_t^\top \boldsymbol{P}_t \boldsymbol{U}_t \boldsymbol{U}_t^\top) + \boldsymbol{R}_t,$$

where
$$\boldsymbol{R}_t = \eta (\boldsymbol{I}_r + \eta \boldsymbol{P}_t) \boldsymbol{U}_t \boldsymbol{J}_t^\top \boldsymbol{J}_t \boldsymbol{U}_t^\top + \eta \boldsymbol{U}_t \boldsymbol{J}_t^\top \boldsymbol{J}_t \boldsymbol{U}_t^\top (\boldsymbol{I}_r + \eta \boldsymbol{P}_t) - \eta^2 \boldsymbol{U}_t (\boldsymbol{J}_t^\top \boldsymbol{J}_t)^2 \boldsymbol{U}_t^\top.$$

By the singular value inequality,
$$\sigma_1(\boldsymbol{P}_{t+1}) \leq ((1 - \eta\Delta/4)^2 + 8\eta^2 \lambda_1^2) \cdot \sigma_1(\boldsymbol{P}_t) + \sigma_1(\boldsymbol{R}_t)$$
$$\leq (1 - \eta\Delta/4) \cdot \sigma_1(\boldsymbol{P}_t) + \sigma_1(\boldsymbol{R}_t),$$

where we use $\Delta/4 \leq \sigma_r^2(\boldsymbol{U}_t) \leq \sigma_1^2(\boldsymbol{U}_t) \leq 2\lambda_1$ in the first inequality and $\eta \leq \frac{\Delta}{36\lambda_1^2}$ in the second inequality. For the remainder term $\boldsymbol{R}_t$, by the singular value inequality and the condition $\eta \leq \frac{\Delta^2}{36\lambda_1^3}$, we have
$$\sigma_1(\boldsymbol{R}_t) \leq \sigma_1^2(\boldsymbol{J}_t) \leq (1 - \eta\Delta/3)^t \cdot \frac{8\lambda_1^2}{\Delta},$$

where the last inequality follows from Lemma 19. Then by deduction, we have
$$\frac{\sigma_1(\boldsymbol{P}_{t+1})}{(1 - \eta\Delta/4)^{t+1}} \leq \frac{\sigma_1(\boldsymbol{P}_t)}{(1 - \eta\Delta/4)^t} + \left(\frac{1 - \eta\Delta/3}{1 - \eta\Delta/4}\right)^t \frac{8\lambda_1^2}{(1 - \eta\Delta/4)\Delta}$$
$$\leq \sigma_1(\boldsymbol{P}_0) + \sum_{i=1}^t \left(\frac{1 - \eta\Delta/3}{1 - \eta\Delta/4}\right)^i \frac{8\lambda_1^2}{(1 - \eta\Delta/4)\Delta}$$
$$\leq \sigma_1(\boldsymbol{P}_0) + \frac{96\lambda_1^2}{\eta\Delta^2} \leq \frac{100\lambda_1^2}{\eta\Delta^2},$$

where the last inequality follows from $\sigma_1(\boldsymbol{P}_0) \leq 2\lambda_1$. Therefore, it takes $T_{\boldsymbol{P}}^\epsilon = \mathcal{O}(\frac{4}{\eta\Delta} \log \frac{100\lambda_1^2}{\eta\Delta^2\epsilon})$ iterations to achieve $\sigma_1(\boldsymbol{P}_t) \leq \epsilon$. ∎

### B.4 PROOF OF THEOREM 2

By combining Lemma 19 and Lemma 20, we can prove Theorem 2.

**Proof** Observe that
$$\|\boldsymbol{\Sigma}_r - \boldsymbol{X}_t \boldsymbol{X}_t^\top\|_\mathrm{F} \leq \|\boldsymbol{P}_t\|_\mathrm{F} + 2\|\boldsymbol{J}_t \boldsymbol{X}_t^\top\|_\mathrm{F} \leq r\sigma_1(\boldsymbol{P}_t) + 2r\sqrt{2\lambda_1}\sigma_1(\boldsymbol{J}_t), \quad \forall \boldsymbol{X}_t \in \mathcal{R},$$

where we use the fact that $\|\boldsymbol{A}\|_\mathrm{F} \leq r\sigma_1(\boldsymbol{A})$ for any rank-$r$ matrix $\boldsymbol{A}$. Let
$$T^\epsilon = \max\left\{T_{\boldsymbol{J}}^{\epsilon^2/(32r^2\lambda_1)}, T_{\boldsymbol{P}}^{\epsilon/(2r)}\right\}.$$

Then, $\|\boldsymbol{\Sigma}_r - \boldsymbol{X}_t \boldsymbol{X}_t^\top\|_\mathrm{F} \leq \epsilon$ for all $t \geq T^\epsilon$. Theorem 2 follows from $T^\epsilon = \mathcal{O}(\frac{6}{\eta\Delta} \log \frac{200r\lambda_1^2}{\eta\Delta^2\epsilon})$. ∎

## C PROOF OF PROPOSITION 3

**Proof** Consider $\boldsymbol{X}$ with $\sigma_1(\boldsymbol{X}) \leq \frac{1}{\sqrt{3\eta}}$. Let $\boldsymbol{X}_t$ be the GD sequence initialized by $\boldsymbol{X}$. By Corollary 2 of Lee et al. [2019], we know GD sequence almost surely avoids the strict saddle points. By Zhu et al. [2021], we know all the saddle points are strict and all the local minima are global minima. Therefore, we conclude that the GD sequence converges to the global minima almost surely.

Now it remains to show that Assumption 4 must hold if the GD sequence converges to the global minima. Indeed, if we suppose Assumption 4 does not hold, then the GD sequence will converge with $\lim_{t\to\infty} \sigma_1(\boldsymbol{u}_{k,t}) = 0$ for some $k \leq r$. This means the GD sequence converges to a saddle point, since any stationary point with some $\boldsymbol{u}_{k,t} = \boldsymbol{0}$ ($k \leq r$) is a saddle point, rather than a global minimum. This leads to the contradiction. ∎

# D ANALYSIS OF LARGE INITIALIZATION

In this section, we will prove Theorem 5 as well as the results in Section 4. Before delving further, we first write down the update rules of $\boldsymbol{u}_{k,t}$ and $\boldsymbol{K}_{k,t}$. Recall that $\boldsymbol{u}_{k,t}$ and $\boldsymbol{K}_{k,t}$ are the $k$-th and $(k+1)$-to-$d$-th rows of $\boldsymbol{X}_t$. The update rules are given by

$$\boldsymbol{u}_{k,t+1} = \boldsymbol{u}_{k,t} + \eta\lambda_k\boldsymbol{u}_{k,t} - \eta\boldsymbol{u}_{k,t}\boldsymbol{X}_t^\top\boldsymbol{X}_t, \tag{24}$$

$$\boldsymbol{K}_{k,t+1} = \boldsymbol{K}_{k,t} + \eta\boldsymbol{\Gamma}_k\boldsymbol{K}_{k,t} - \eta\boldsymbol{K}_{k,t}\boldsymbol{X}_t^\top\boldsymbol{X}_t, \tag{25}$$

where $\boldsymbol{\Gamma}_k = \mathrm{diag}(\lambda_{k+1}, \dots, \lambda_d)$. We also remind readers that $\boldsymbol{u}_{k,t} \in \mathbb{R}^{1\times r}$ is a row vector. Moreover, we let $\boldsymbol{\Pi}_{\boldsymbol{u}_{k,t}}$ denote the projection matrix associated with $\boldsymbol{u}_{k,t}$, that is,

$$\boldsymbol{\Pi}_{\boldsymbol{u}_{k,t}} = \boldsymbol{u}_{k,t}^\top(\boldsymbol{u}_{k,t}\boldsymbol{u}_{k,t}^\top)^{-1}\boldsymbol{u}_{k,t} \in \mathbb{R}^{r\times r}.$$

Also, we let $\boldsymbol{G}_{k,t}$ denote the first $k$ rows of $\boldsymbol{X}_t$.

## D.1 PROOFS FOR SECTION 4

In this section, we collect proofs related to the rank-two matrix approximation.

### D.1.1 Proof of Lemma 8

**Proof** Note that $t_{\mathrm{init},1} \le T_1 + T_{\boldsymbol{K}}$, where

$$T_1 = \min\{t \ge 0 \mid \sigma_1^2(\boldsymbol{X}_t) \le 2\lambda_1\}$$

is the first time when $\sigma_1^2(\boldsymbol{X}_t)$ is smaller than $2\lambda_1$, and

$$T_{\boldsymbol{K}} = \min\{t \ge 0 \mid \sigma_1^2(\boldsymbol{K}_{k,t+T_1}) \le \lambda_k - \frac{3\Delta}{4}, \forall k \le r\}.$$

To prove the lemma, it suffices to analyze $T_1$ and $T_{\boldsymbol{K}}$ separately.

First, we analyze $T_1$ as follows.

- If $\sigma_1^2(\boldsymbol{X}_0) \le 2\lambda_1$, then $T_1 = 0$.
- If $2\lambda_1 < \sigma_1^2(\boldsymbol{X}_0) < 1/(3\eta)$, then by Lemma 13, $\sigma_1^2(\boldsymbol{X}_t) \le 1/(3\eta)$ for all $t$. Furthermore, it follows from Lemma 16 that

$$\sigma_1(\boldsymbol{X}_{t+1}) \le (1 + \eta\lambda_1 - \eta\sigma_1^2(\boldsymbol{X}_t)) \cdot \sigma_1(\boldsymbol{X}_t)$$
$$\le (1 - \eta\lambda_1) \cdot \sigma_1(\boldsymbol{X}_t), \quad \forall t < T_1,$$

  where the second inequality uses $\sigma_1^2(\boldsymbol{X}_t) > 2\lambda_1$ for all $t < T_1$. It implies that

$$\sigma_1(\boldsymbol{X}_t) \le (1 - \eta\lambda_1)^t \cdot \sigma_1(\boldsymbol{X}_0)$$

  for all $t \le T_1$ and

$$T_1 = \mathcal{O}\left(\frac{1}{\eta\lambda_1}\log\frac{\sigma_1(\boldsymbol{X}_0)}{\sqrt{2\lambda_1}}\right).$$

By Lemma 13, we have $\sigma_1^2(\boldsymbol{X}_t) \le 2\lambda_1$ for all $t \ge T_1$.

Next, we analyze $T_{\boldsymbol{K}}$ and the following quantities

$$T_{\boldsymbol{K}_k} = \min\{t \ge 0 \mid \sigma_1^2(\boldsymbol{K}_{k,t+T_1}) \le \lambda_k - \frac{3\Delta}{4}\}.$$

Recall that $K_{k,t}$ is the $(k+1)$-to-$d$-th rows of $X_t$. Then by (25), we have

$$
\begin{aligned}
K_{k,t+1} &= K_{k,t} + \eta \Gamma_k K_{k,t} - \eta K_{k,t} X_t^\top X_t \\
&= \underbrace{\frac{1}{2} K_{k,t} - \eta K_{k,t} K_{k,t}^\top K_{k,t}}_{B} + \underbrace{(\frac{1}{4} I_{d-k} + \eta \Gamma_k) K_{k,t}}_{C} + \underbrace{K_{k,t}(\frac{1}{4} I_k - \eta G_{k,t}^\top G_{k,t})}_{D},
\end{aligned}
$$

where $\Gamma_k = \mathrm{diag}(\lambda_{k+1}, \ldots, \lambda_d)$ and $G_{k,t} \in \mathbb{R}^{k \times r}$ is the first $k$ rows of $X_t$. By the singular value inequality, we obtain

$$
\sigma_1(K_{k,t+1}) \leq \sigma_1(B) + \sigma_1(C) + \sigma_1(D).
$$

For the first term $B$, similar to Lemma 17, we can show that

$$
\sigma_1(B) = \sigma_1(K_{k,t})/2 - \eta \sigma_1^3(K_{k,t}), \quad \forall t \geq T_1.
$$

For the second term $C$, by the singular value inequality,

$$
\sigma_1(C) \leq (\frac{1}{4} + \eta \lambda_{k+1}) \cdot \sigma_1(K_{k,t}).
$$

For the third term $D$, since $G_{k,t}^\top G_{k,t}$ is PSD and $\eta \sigma_1^2(G_{k,t}) \leq \frac{1}{4}$ for all $t \geq T_1$, we have

$$
\sigma_1(D) \leq \sigma_1(K_{k,t})/4, \quad \forall t \geq T_1.
$$

Combining,

$$
\sigma_1(K_{k,t+1}) \leq (1 + \eta \lambda_{k+1} - \eta \sigma_1^2(K_{k,t})) \cdot \sigma_1(K_{k,t}), \quad \forall t \geq T_1, \quad \forall k \leq r. \tag{26}
$$

Since $\lambda_{k+1} \leq \lambda_k - \Delta$ for $k \leq r$, (26) implies that

$$
\sigma_1(K_{k,t+T_1+1}) \leq (1 - \eta \Delta/4) \cdot \sigma_1(K_{k,t+T_1}), \quad \forall t < T_{K_k}, \quad \forall k \leq r.
$$

Hence, $\sigma_1(K_{k,t+T_1}) \leq (1 - \eta \Delta/4)^t \cdot \sigma_1(K_{k,T_1})$ for all $t \leq T_{K_k}$. In particular,

$$
T_{K_k} = \mathcal{O}\left( \frac{2}{\eta \Delta} \log \frac{\sigma_1^2(K_{k,T_1})}{\lambda_k - \frac{3\Delta}{4}} \right) \quad \text{and} \quad T_K = \mathcal{O}\left( \frac{2}{\eta \Delta} \log \frac{8\lambda_1}{\Delta} \right),
$$

where we use $\sigma_1^2(K_{k,T_1}) \leq 2\lambda_1$ and $\lambda_k - \frac{3\Delta}{4} \geq \frac{\Delta}{4}$.

Finally, similar to Lemma 13 and 14, for any $a \geq \lambda_{k+1}$, if $\sigma_1^2(K_{k,t+T_1}) \leq a$, then $\sigma_1^2(K_{k,t'+T_1}) \leq a$ for all $t' \geq t$. This implies that $\sigma_1^2(K_{k,t+T_1}) \leq \lambda_k - \frac{3\Delta}{4}$ for all $t \geq T_K$ for $k \leq r$. ∎

### D.1.2 Proof of Lemma 9

**Proof** This lemma is a special case of Lemma 21, where we take $k = 1$ and $t_{\mathrm{init}} = t_{\mathrm{init},1}$. Notice that $G_{0,t} = 0$ and $X_t \in \mathcal{S}$ for all $t \geq t_{\mathrm{init},1}$ by Lemma 8. Thus, the conditions in Lemma 21 trivially hold. Then Lemma 9 immediately follows from Lemma 21. ∎

### D.1.3 Proof of Lemma 10

**Proof** The lemma is a special case of Lemma 22 and Lemma 23. In Lemma 22, we take $k = 1$ and $t_{\mathrm{init}} = t_{\mathrm{init},1} + T_{u_1}$. In Lemma 23, we take $k = 1$ and $t_{\mathrm{init}} = t_{\mathrm{init},1} + T_{u_1} + t^*$. ∎

### D.1.4 Proof of Lemma 11

**Proof** This lemma is a special case of Lemma 21, where we take $k = 2$ and $t_{\mathrm{init}} = t_1 + t_1^*$. ∎

## D.2 PROOF OF THEOREM 5

**Proof** To prove this theorem, we will use an inductive argument. Our induction hypotheses are listed below:

- H($k$, 1). $\sigma_1(\boldsymbol{u}_{k,t}\boldsymbol{G}_{k-1,t}^{\top}) \leq \sqrt{\frac{\Delta}{8}}\min\{\sigma_1(\boldsymbol{u}_{k,t_{\text{init},k}}), \sqrt{\frac{\Delta}{2}}\} \cdot (1-\eta\Delta/6)^{t-t_{\text{init},k}}$ for all $t \geq t_{\text{init},k}$.

- H($k$, 2). $T_{\boldsymbol{u}_k} = \mathcal{O}\left(\frac{4}{\eta\Delta}\log\frac{\Delta}{2\sigma_1^2(\boldsymbol{u}_{k,t_{\text{init},k}})}\right)$ and $\sigma_1^2(\boldsymbol{u}_{k,t}) \geq \frac{\Delta}{2}$ for all $t \geq t_{\text{init},k} + T_{\boldsymbol{u}_k}$.

- H($k$, 3). $\sigma_1(\boldsymbol{u}_{k,t}\boldsymbol{K}_{k,t}^{\top}) \leq (1-\eta\Delta/6)^{t-t_k}$ for all $t \geq t_k$.

Note that H(1,1) trivially holds because $\boldsymbol{G}_{0,t} = \boldsymbol{0}$. Then we prove H($k$, 1), H($k$, 2), H($k$, 3), H($k+1, 1$) successively until H($r$, 3).

- $\{\text{H}(j,\cdot)\}_{j<k} + \text{H}(k,1) \rightarrow \text{H}(k,2)$

  This follows from Lemma 21, where we take $t_{\text{init}} = t_{\text{init},k}$.

- $\{\text{H}(j,\cdot)\}_{j<k} + \text{H}(k,1) + \text{H}(k,2) \rightarrow \text{H}(k,3)$

  This follows from Lemma 22, where we take $t_{\text{init}} = t_{\text{init},k} + T_{\boldsymbol{u}_k}$.

- $\{\text{H}(j,\cdot)\}_{j\leq k} \rightarrow \text{H}(k+1,1)$

  By $\{\text{H}(j,3)\}_{j\leq k}$,

  $$\sigma_1(\boldsymbol{u}_{k+1,t}\boldsymbol{G}_{k,t}^{\top}) \leq \sum_{j\leq k}\sigma_1(\boldsymbol{u}_{j,t}\boldsymbol{K}_{j,t}^{\top}) \leq r(1-\eta\Delta/6)^{t-t_k},$$

  for all $t \geq t_k$. By definition of $t_k^*$, we have

  $$r(1-\eta\Delta/6)^{t_k^*} \leq \sqrt{\frac{\Delta}{8}}\min\{\sigma_1(\boldsymbol{u}_{k,t_k+t_k^*}), \sqrt{\frac{\Delta}{2}}\}.$$

  Then H($k+1, 1$) follows from the definition $t_{\text{init},k+1} = t_k + t_k^*$.

By induction, H($k$, $\cdot$) holds for all $k \leq r$.

For all $t \geq t_k$, (18) follows from Lemma 23, where $t_{\text{init}}$ is taken as $t_k$.

For all $t \geq t_{\text{init},r} + T_{\boldsymbol{u}_r}$, we have $\sigma_1^2(\boldsymbol{u}_{k,t}) \geq \frac{\Delta}{2}$ for all $k \leq r$. Simultaneously,

$$\sum_{j\leq r}\sigma_1(\boldsymbol{u}_{j,t}\boldsymbol{K}_{j,t}^{\top}) \leq r(1-\eta\Delta/6)^{t-(t_{\text{init},r}+T_{\boldsymbol{u}_r}+t^*)}$$

holds for all $t \geq t_{\text{init},r} + T_{\boldsymbol{u}_r} + t^*$. Let $\boldsymbol{U}_t$ be the first $r$ rows of $\boldsymbol{X}_t$. Viewing $\boldsymbol{U}_t\boldsymbol{U}_t^{\top}$ as the sum of diagonal elements and off-diagonal elements, we find that

$$\sigma_r^2(\boldsymbol{U}_t) \geq \Delta/2 - r(1-\eta\Delta/6)^{t-(t_{\text{init},r}+T_{\boldsymbol{u}_r}+t^*)}$$

for all $t \geq t_{\text{init},r} + T_{\boldsymbol{u}_r} + t^*$. Hence, $\sigma_r^2(\boldsymbol{U}_t) \geq \Delta/4$ for all $t \geq t_{\text{init},r} + T_{\boldsymbol{u}_r} + t^* + t^{\sharp}$, where

$$t^{\sharp} = \frac{\log(\Delta/(4r))}{\log(1-\eta\Delta/6)}.$$

This implies that $\boldsymbol{X}_t \in \mathcal{R}$ for $t \geq t_{\mathcal{R}} := t_{\text{init},r} + T_{\boldsymbol{u}_r} + t^* + t^{\sharp}$.

The property (2) is merely an application of Theorem 2. ∎

## D.3 PROOF OF THEOREM 7

**Proof** This property immediately follows from Theorem 5. ∎

## D.4 TECHNICAL LEMMAS

This section collects technical lemmas that are used in previous sections. Let us recall that $\boldsymbol{u}_{k,t}$ and $\boldsymbol{K}_{k,t}$ are the $k$-th and the $(k+1)$-to-$d$-th rows of $\boldsymbol{X}_t$ respectively. The projection matrix associated with $\boldsymbol{u}_{k,t}$ is denoted by

$$\boldsymbol{\Pi}_{\boldsymbol{u}_k,t} = \boldsymbol{u}_{k,t}^\top (\boldsymbol{u}_{k,t} \boldsymbol{u}_{k,t}^\top)^{-1} \boldsymbol{u}_{k,t}.$$

The first $k$ rows of $\boldsymbol{X}_t$ are denoted by $\boldsymbol{G}_{k,t}$, and $\boldsymbol{G}_{0,t} = \boldsymbol{0}$ by definition.

### D.4.1 Dynamics

This subsection contains lemmas describing the dynamics of the GD sequence.

Lemma 21 shows that when $\sigma_1(\boldsymbol{u}_{k,t} \boldsymbol{G}_{k-1,t}^\top)$ is sufficiently small, the signal term $\sigma_1^2(\boldsymbol{u}_{k,t+1})$ can rise above $\Delta/2$ quickly. Moreover, as shown in Lemma 21, the term $\sigma_1^2(\boldsymbol{u}_{k,t+1})$ will remain larger than $\Delta/2$.

**Lemma 21** *Suppose $\eta \leq \frac{1}{12\lambda_1}$, $\boldsymbol{X}_t \in \mathcal{S}$, and for some $t_{\mathrm{init}} \geq 0$ and $k \leq r$, the condition*

$$\sigma_1(\boldsymbol{u}_{k,t} \boldsymbol{G}_{k-1,t}^\top) \leq \sqrt{\frac{\Delta}{8}} \min\{\sigma_1(\boldsymbol{u}_{k,t_{\mathrm{init}}}), \sqrt{\frac{\Delta}{2}}\} \cdot (1 - \eta\Delta/6)^{t-t_{\mathrm{init}}}$$

*holds for all $t \geq t_{\mathrm{init}}$. Then $\sigma_1^2(\boldsymbol{u}_{k,t}) \geq \frac{\Delta}{2}$ for all $t \geq t_{\mathrm{init}} + T_{\boldsymbol{u}_k}$, where*

$$T_{\boldsymbol{u}_k} = \mathcal{O}\left(\frac{4}{\eta\Delta} \log \frac{\Delta}{2\sigma_1^2(\boldsymbol{u}_{k,t_{\mathrm{init}}})}\right).$$

*In addition, for all $t \geq t_{\mathrm{init}}$, we have*

$$\sigma_1^2(\boldsymbol{u}_{k,t+1}) \geq (1 + 2\eta\lambda_k - \eta\Delta/4 - 2\eta\sigma_1^2(\boldsymbol{u}_{k,t}) - 2\eta\sigma_1^2(\boldsymbol{K}_{k,t}\boldsymbol{\Pi}_{\boldsymbol{u}_k,t}))) \cdot \sigma_1^2(\boldsymbol{u}_{k,t}), \tag{27}$$

*where $\boldsymbol{\Pi}_{\boldsymbol{u}_k,t} = \boldsymbol{u}_{k,t}^\top (\boldsymbol{u}_{k,t} \boldsymbol{u}_{k,t}^\top)^{-1} \boldsymbol{u}_{k,t}$ is the projection matrix associated with $\boldsymbol{u}_{k,t}$.*

**Proof** First, we show that $\sigma_1^2(\boldsymbol{u}_{k,t}) \geq \min\{\sigma_1^2(\boldsymbol{u}_{k,t_{\mathrm{init}}}), \frac{\Delta}{2}\}$ for all $t \geq t_{\mathrm{init}}$ by induction.

This is true when $t = t_{\mathrm{init}}$. Now suppose $\sigma_1^2(\boldsymbol{u}_{k,t}) \geq \min\{\sigma_1^2(\boldsymbol{u}_{k,t_{\mathrm{init}}}), \frac{\Delta}{2}\}$ for some $t \geq t_{\mathrm{init}}$. By assumption, $\sigma_1^2(\boldsymbol{u}_{k,t} \boldsymbol{G}_{k-1,t}^\top) \leq \frac{\Delta}{8} \min\{\sigma_1^2(\boldsymbol{u}_{k,t_{\mathrm{init}}}), \frac{\Delta}{2}\} \leq \frac{\Delta}{8}\sigma_1^2(\boldsymbol{u}_{k,t})$. Then by Lemma 24 and $\boldsymbol{X}_t \in \mathcal{S}$, we have

$$\sigma_1^2(\boldsymbol{u}_{k,t+1}) \geq (1 + 2\eta\lambda_k - 2\eta\sigma_1^2(\boldsymbol{u}_{k,t}) - 2\eta\sigma_1^2(\boldsymbol{K}_{k,t}\boldsymbol{\Pi}_{\boldsymbol{u}_k,t}))) \cdot \sigma_1^2(\boldsymbol{u}_{k,t}) - \frac{\eta\Delta}{4}\sigma_1^2(\boldsymbol{u}_{k,t}) \tag{28}$$

$$\geq (1 + 5\eta\Delta/4 - 2\eta\sigma_1^2(\boldsymbol{u}_{k,t})) \cdot \sigma_1^2(\boldsymbol{u}_{k,t}). \tag{29}$$

Then we consider two cases.

- If $\sigma_1^2(\boldsymbol{u}_{k,t}) \leq \frac{5\Delta}{8}$, then $\sigma_1^2(\boldsymbol{u}_{k,t+1}) \geq \sigma_1^2(\boldsymbol{u}_{k,t}) \geq \min\{\sigma_1^2(\boldsymbol{u}_{k,t_{\mathrm{init}}}), \frac{\Delta}{2}\}$.
- If $\sigma_1^2(\boldsymbol{u}_{k,t}) \geq \frac{5\Delta}{8}$, then

$$\sigma_1^2(\boldsymbol{u}_{k,t+1}) \geq (1 + \frac{5\eta\Delta}{4} - \frac{5\eta\Delta}{4}) \cdot \frac{5\Delta}{8} = \frac{5\Delta}{8}$$

$$\geq \min\{\sigma_1^2(\boldsymbol{u}_{k,t_{\mathrm{init}}}), \frac{\Delta}{2}\},$$

where the first inequality uses the fact that $g_8(s) = (1 + \frac{5\eta\Delta}{4} - 2\eta s)s$ is increasing on $(-\infty, 1/4\eta]$.

In both cases, we have $\sigma_1^2(\boldsymbol{u}_{k,t+1}) \geq \min\{\sigma_1^2(\boldsymbol{u}_{k,\mathrm{init}}), \frac{\Delta}{2}\}$. The claim then follows by induction.

Furthermore, the above analysis shows that inequalities 28 and 29 hold for all $t \geq t_{\mathrm{init}}$, which leads to the inequality 27.

Let

$$T_{\boldsymbol{u}_k} = \min\{t \geq 0 \mid \sigma_1^2(\boldsymbol{u}_{k,t+t_{\text{init}}}) \geq \frac{\Delta}{2}\}.$$

Then for $t < T_{\boldsymbol{u}_k}$, we have $\sigma_1^2(\boldsymbol{u}_{k,t+t_{\text{init}}}) < \frac{\Delta}{2}$ and by inequality 29,

$$\sigma_1^2(\boldsymbol{u}_{k,t+1+t_{\text{init}}}) \geq (1 + \eta\Delta/4) \cdot \sigma_1^2(\boldsymbol{u}_{k,t+t_{\text{init}}}).$$

Hence, for all $t \leq T_{\boldsymbol{u}_k}$, we have

$$\sigma_1^2(\boldsymbol{u}_{k,t+t_{\text{init}}}) \geq (1 + \eta\Delta/4)^t \cdot \sigma_1^2(\boldsymbol{u}_{k,t_{\text{init}}}),$$

and

$$T_{\boldsymbol{u}_k} = \mathcal{O}\left(\frac{4}{\eta\Delta} \log \frac{\Delta}{2\sigma_1^2(\boldsymbol{u}_{k,t_{\text{init}}})}\right).$$

Finally, by inequality 29, we have for any $a \leq \frac{5\Delta}{8}$, if $\sigma_1^2(\boldsymbol{u}_{k,t}) \geq a$, then $\sigma_1^2(\boldsymbol{u}_{k,t+1}) \geq a$. Thus, by induction, $\sigma_1^2(\boldsymbol{u}_{k,t}) \geq \frac{\Delta}{2}$ for all $t \geq t_{\text{init}} + T_{\boldsymbol{u}_k}$. ∎

Lemma 22 shows that when the noise terms $\sigma_1(\boldsymbol{u}_{j,t}\boldsymbol{K}_{j,t}^\top)$ converge linearly to zero for all $j < k$ and the $k$-th signal term $\sigma_1^2(\boldsymbol{u}_{k,t}) \geq \frac{\Delta}{2}$, the noise term $\sigma_1(\boldsymbol{u}_{k,t}\boldsymbol{K}_{k,t}^\top)$ will also converge linearly to zero. The key component is to analyze the SNR

$$\frac{\sigma_1^2(\boldsymbol{u}_{k,t})}{\sigma_1(\boldsymbol{u}_{k,t}\boldsymbol{K}_{k,t}^\top)}.$$

**Lemma 22** *Suppose $\eta \leq \frac{\Delta}{100\lambda_1^2}$, $\boldsymbol{X}_t \in \mathcal{S}$, and for some $t_{\text{init}} \geq 0$ and $k \leq r$, the conditions*

$$\sigma_1(\boldsymbol{u}_{j,t}\boldsymbol{K}_{j,t}^\top) \leq (1 - \eta\Delta/6)^{t-t_{\text{init}}}, \quad \forall j < k, \tag{30}$$

$$\sigma_1(\boldsymbol{u}_{k,t}\boldsymbol{G}_{k-1,t}^\top) \leq \frac{\Delta}{4}(1 - \eta\Delta/6)^{t-t_{\text{init}}}, \tag{31}$$

$$\sigma_1^2(\boldsymbol{u}_{k,t}) \geq \frac{\Delta}{2} \tag{32}$$

*hold for all $t \geq t_{\text{init}}$. Then we have*

$$\sigma_1(\boldsymbol{u}_{k,t}\boldsymbol{K}_{k,t}^\top) \leq (1 - \eta\Delta/6)^{t-t_{\text{init}}-t^*}$$

*for all $t \geq t_{\text{init}} + t^*$, where*

$$t^* = \log\left(\frac{\Delta^2}{8\lambda_1^3 + 144r^2\lambda_1}\right) / \log(1 - \eta\Delta/6).$$

**Proof** By condition 31, we can apply Lemma 21 to obtain

$$\sigma_1^2(\boldsymbol{u}_{k,t+1}) \geq (1 + 2\eta\lambda_k - \eta\Delta/4 - 2\eta\sigma_1^2(\boldsymbol{u}_{k,t}) - 2\eta\sigma_1^2(\boldsymbol{K}_{k,t}\boldsymbol{\Pi}_{\boldsymbol{u}_k,t})) \cdot \sigma_1^2(\boldsymbol{u}_{k,t})$$

for all $t \geq t_{\text{init}}$. By Lemma 25, we have

$$\begin{aligned}
&\sigma_1(\boldsymbol{u}_{k,t+1}\boldsymbol{K}_{k,t+1}^\top)\\
&\leq (1 + \eta\lambda_k + \eta\lambda_{k+1} - 2\eta\sigma_1^2(\boldsymbol{u}_{k,t}) - 2\eta\sigma_1^2(\boldsymbol{K}_{k,t}\boldsymbol{\Pi}_{\boldsymbol{u}_k,t}) + 25\eta^2\lambda_1^2) \cdot \sigma_1(\boldsymbol{u}_{k,t}\boldsymbol{K}_{k,t}^\top)\\
&\quad + 3\eta\sigma_1(\boldsymbol{u}_{k,t}\boldsymbol{G}_{k-1,t}^\top)\sigma_1(\boldsymbol{K}_{k,t}\boldsymbol{G}_{k-1,t}^\top)
\end{aligned}$$

for all $t \geq t_{\text{init}}$. Divide both sides of the inequality by $\sigma_1^2(\boldsymbol{u}_{k,t+1})$. By Lemma 26 and $\sigma_1^2(\boldsymbol{u}_{k,t+1}) \geq \frac{\Delta}{2}$, we have

$$\frac{\sigma_1(\boldsymbol{u}_{k,t+1}\boldsymbol{K}_{k,t+1}^\top)}{\sigma_1^2(\boldsymbol{u}_{k,t+1})} \leq (1 - \eta\Delta/6)\frac{\sigma_1(\boldsymbol{u}_{k,t}\boldsymbol{K}_{k,t}^\top)}{\sigma_1^2(\boldsymbol{u}_{k,t})} + \frac{6\eta}{\Delta}\sigma_1(\boldsymbol{u}_{k,t}\boldsymbol{G}_{k-1,t}^\top)\sigma_1(\boldsymbol{K}_{k,t}\boldsymbol{G}_{k-1,t}^\top) \tag{33}$$

for all $t \geq t_{\text{init}}$. Observe that by condition 30 and definitions of $\boldsymbol{u}_{k,t}$, $\boldsymbol{K}_{k,t}$, and $\boldsymbol{G}_{k-1,t}$, we have

$$\max\{\sigma_1(\boldsymbol{u}_{k,t}\boldsymbol{G}_{k-1,t}^\top), \sigma_1(\boldsymbol{K}_{k,t}\boldsymbol{G}_{k-1,t}^\top)\} \leq \sum_{j<k} \sigma_1(\boldsymbol{u}_{k,t}\boldsymbol{K}_{k,t}^\top) \leq r(1 - \eta\Delta/6)^{t-t_{\text{init}}} \tag{34}$$

for all $t \geq t_{\text{init}}$. Combining (33) and (34),

$$\frac{\sigma_1(\boldsymbol{u}_{k,t+1}\boldsymbol{K}_{k,t+1}^\top)}{\sigma_1^2(\boldsymbol{u}_{k,t+1})} \leq (1 - \eta\Delta/6)\frac{\sigma_1(\boldsymbol{u}_{k,t}\boldsymbol{K}_{k,t}^\top)}{\sigma_1^2(\boldsymbol{u}_{k,t})} + \frac{6\eta r^2}{\Delta}(1 - \eta\Delta/6)^{2(t-t_{\text{init}})}$$

for all $t \geq t_{\text{init}}$. Therefore, for all $t \geq t_{\text{init}}$,

$$Q_{t+1} \leq (1 - \eta\Delta/6) \cdot Q_t,$$

where the quantity $Q_t$ is given by

$$Q_t = \frac{\sigma_1(\boldsymbol{u}_{k,t}\boldsymbol{K}_{k,t}^\top)}{\sigma_1^2(\boldsymbol{u}_{k,t})} + \frac{36r^2}{\Delta^2}(1 - \eta\Delta/6)^{2(t-t_{\text{init}})-1}.$$

By induction, we have

$$\frac{\sigma_1(\boldsymbol{u}_{k,t}\boldsymbol{K}_{k,t}^\top)}{\sigma_1^2(\boldsymbol{u}_{k,t})} \leq (1 - \eta\Delta/6)^{t-t_{\text{init}}}\left(\frac{\sigma_1(\boldsymbol{u}_{k,t_{\text{init}}}\boldsymbol{K}_{k,t_{\text{init}}}^\top)}{\sigma_1^2(\boldsymbol{u}_{k,t_{\text{init}}})} + \frac{36r^2}{\Delta^2}(1 - \eta\Delta/6)^{-1}\right).$$

This implies that

$$\sigma_1(\boldsymbol{u}_{k,t}\boldsymbol{K}_{k,t}^\top) \leq \frac{8\lambda_1^3 + 144r^2\lambda_1}{\Delta^2} \cdot (1 - \eta\Delta/6)^{t-t_{\text{init}}},$$

where we use $1 - \eta\Delta/6 \geq 1/2$, $\sigma_1^2(\boldsymbol{X}_t) \leq 2\lambda_1$, and $\sigma_1^2(\boldsymbol{u}_{k,t}) \geq \frac{\Delta}{2}$ for all $t \geq t_{\text{init}}$. By definition of $t^*$, we have

$$(1 - \eta\Delta/6)^{t^*} \leq \frac{\Delta^2}{8\lambda_1^3 + 144r^2\lambda_1}.$$

Thus, for all $t \geq t_{\text{init}} + t^*$, we have

$$\sigma_1(\boldsymbol{u}_{k,t}\boldsymbol{K}_{k,t}^\top) \leq (1 - \eta\Delta/6)^{t-t_{\text{init}}-t^*},$$

which concludes the proof. ∎

Let $p_{k,t} = \lambda_k - \sigma_1^2(\boldsymbol{u}_{k,t})$ be the error term associated with the $k$-th signal. Lemma 23 shows that when the noise terms $\sigma_1(\boldsymbol{u}_{j,t}\boldsymbol{K}_{j,t}^\top)$ converge linearly to zero for all $j \leq k$ and the $k$-th signal term $\sigma_1^2(\boldsymbol{u}_{k,t}) \geq \frac{\Delta}{2}$, this signal term will converge fast to $\lambda_k$. Specifically, the error term $|p_{k,t}|$ will converge to zero at a linear rate. The analysis is similar to Lemma 20.

**Lemma 23** *Suppose* $\eta \leq \frac{\Delta}{100\lambda_1^2}$, $\boldsymbol{X}_t \in \mathcal{S}$, *and for some* $t_{\text{init}} \geq 0$ *and* $k \leq r$, *the conditions*

$$\sigma_1(\boldsymbol{u}_{j,t}\boldsymbol{K}_{j,t}^\top) \leq (1 - \eta\Delta/6)^{t-t_{\text{init}}}, \quad \forall j \leq k, \tag{35}$$

$$\sigma_1^2(\boldsymbol{u}_{k,t}) \geq \frac{\Delta}{2} \tag{36}$$

*hold for all* $t \geq t_{\text{init}}$. *Then for all* $t \geq t_{\text{init}}$, *we have*

$$|p_{k,t}| \leq (2\lambda_1 + \frac{24r}{\eta\Delta}) \cdot (1 - \eta\Delta/8)^{t-t_{\text{init}}},$$

*where* $p_{k,t} = \lambda_k - \sigma_1^2(\boldsymbol{u}_{k,t})$.

**Proof** Using the notation of $p_{k,t}$, (24) can be rewritten as

$$u_{k,t+1} = u_{k,t} + \eta p_{k,t} u_{k,t} - \eta u_{k,t} W_t.$$

where

$$W_t = G_{k-1,t}^\top G_{k-1,t} + K_{k,t}^\top K_{k,t}.$$

By direction calculation, we have

$$p_{k,t+1} = p_{k,t} \cdot ((1 - \eta\sigma_1^2(u_{k,t}))^2 + \eta^2\lambda_k\sigma_1^2(u_{k,t})) + \text{res}_t$$

where

$$\text{res}_t = 2\eta(1 + \eta p_{k,t}) u_{k,t} W_t u_{k,t}^\top - \eta^2 u_{k,t} W_t^2 u_{k,t}^\top.$$

By the singular value inequality, for all $t \geq t_{\text{init}}$, we have

$$
\begin{aligned}
|p_{k,t+1}| &\leq |p_{k,t}| \cdot ((1 - \eta\sigma_1^2(u_{k,t}))^2 + \eta^2\lambda_k\sigma_1^2(u_{k,t})) + |\text{res}_t| \\
&\leq |p_{k,t}| \cdot ((1 - \eta\Delta/2)^2 + 2\eta^2\lambda_1^2) + |\text{res}_t| \\
&\leq |p_{k,t}| \cdot (1 - \eta\Delta/2) + |\text{res}_t|,
\end{aligned}
\tag{37}
$$

where the second inequality uses $\Delta/2 \leq \sigma_1^2(u_{k,t}) \leq 2\lambda_1$ and the third inequality use $\eta \leq \frac{\Delta}{100\lambda_1^2}$. Using $\eta \leq \frac{\Delta}{100\lambda_1^2}$ and $\sigma_1^2(X_t) \leq 2\lambda_1$, we have

$$|\text{res}_t| \leq \sum_{j \leq k} \sigma_1(u_{k,t} K_{k,t}^\top) \leq r(1 - \eta\Delta/6)^{t - t_{\text{init}}}$$

for all $t \geq t_{\text{init}}$. Substituting this into (37), we obtain

$$
\begin{aligned}
|p_{k,t+1}| &\leq |p_{k,t}| \cdot (1 - \eta\Delta/2) + r(1 - \eta\Delta/6)^{t - t_{\text{init}}} \\
&\leq |p_{k,t}| \cdot (1 - \eta\Delta/8) + r(1 - \eta\Delta/6)^{t - t_{\text{init}}}.
\end{aligned}
$$

This implies that for all $t \geq t_{\text{init}}$,

$$Q_{t+1} \leq Q_t + \frac{r}{1 - \eta\Delta/8}\left(\frac{1 - \eta\Delta/6}{1 - \eta\Delta/8}\right)^{t - t_{\text{init}}},$$

where

$$Q_t = \frac{|p_{k,t}|}{(1 - \eta\Delta/8)^{t - t_{\text{init}}}}.$$

By induction, for all $t \geq t_{\text{init}}$, we have

$$
\begin{aligned}
Q_t &\leq Q_{t_{\text{init}}} + \frac{r}{1 - \eta\Delta/8} \sum_{i=0}^{t-1-t_{\text{init}}}\left(\frac{1 - \eta\Delta/6}{1 - \eta\Delta/8}\right)^i \\
&\leq |p_{k,t_{\text{init}}}| + \frac{24r}{\eta\Delta} \\
&\leq 2\lambda_1 + \frac{24r}{\eta\Delta}.
\end{aligned}
$$

Hence, for all $t \geq t_{\text{init}}$, we have

$$|p_{k,t}| \leq \left(2\lambda_1 + \frac{24r}{\eta\Delta}\right) \cdot (1 - \eta\Delta/8)^{t - t_{\text{init}}},$$

which concludes the proof. $\blacksquare$

### D.4.2 Technical calculations

The following lemmas provide calculations related to an SNR argument, where the SNR refers to the ratio

$$\frac{\sigma_1^2(\boldsymbol{u}_{k,t})}{\sigma_1(\boldsymbol{u}_{k,t}\boldsymbol{K}_{k,t}^\top)}.$$

Recall that $\boldsymbol{u}_{k,t}$ is the $k$-th row of $\boldsymbol{X}_t$ and $\boldsymbol{K}_{k,t}$ represents the $(k+1)$-to-$d$-th rows of $\boldsymbol{X}_t$. Moreover, we recall that

$$\boldsymbol{\Pi}_{\boldsymbol{u}_k,t} = \boldsymbol{u}_{k,t}^\top(\boldsymbol{u}_{k,t}\boldsymbol{u}_{k,t}^\top)^{-1}\boldsymbol{u}_{k,t}$$

is the projection matrix associated with $\boldsymbol{u}_{k,t}$. $\boldsymbol{G}_{k,t}$ collects the first $k$ rows of $\boldsymbol{X}_t$.

Lemma 24 provides a lower bound on $\sigma_1^2(\boldsymbol{u}_{k,t+1})$ in terms of the preceding iteration.

**Lemma 24** *For any $k$ and $t \geq 0$, we have*

$$\sigma_1^2(\boldsymbol{u}_{k,t+1}) \geq (1 + 2\eta\lambda_k - 2\eta\sigma_1^2(\boldsymbol{u}_{k,t}) - 2\eta\sigma_1^2(\boldsymbol{K}_{k,t}\boldsymbol{\Pi}_{\boldsymbol{u}_k,t})) \cdot \sigma_1^2(\boldsymbol{u}_{k,t}) - 2\eta\sigma_1^2(\boldsymbol{u}_{k,t}\boldsymbol{G}_{k-1,t}^\top).$$

**Proof** Substituting (24) into $\sigma_1^2(\boldsymbol{u}_{k,t+1})$ gives that

$$
\begin{aligned}
\sigma_1^2(\boldsymbol{u}_{k,t+1}) &= \boldsymbol{u}_{k,t+1}\boldsymbol{u}_{k,t+1}^\top \\
&= \boldsymbol{u}_{k,t}(\boldsymbol{I}_r + \eta\lambda_k\boldsymbol{I}_r - \eta\boldsymbol{X}_t^\top\boldsymbol{X}_t)^2\boldsymbol{u}_{k,t}^\top \\
&= \boldsymbol{u}_{k,t}(\boldsymbol{I}_r + 2\eta\lambda_k\boldsymbol{I}_r - 2\eta\boldsymbol{X}_t^\top\boldsymbol{X}_t)\boldsymbol{u}_{k,t}^\top + \eta^2\boldsymbol{R}_{k,t} \\
&= \boldsymbol{u}_{k,t}(\boldsymbol{I}_r + 2\eta\lambda_k\boldsymbol{I}_r - 2\eta\sigma_1^2(\boldsymbol{u}_{k,t})\boldsymbol{I}_r - 2\eta\sigma_1^2(\boldsymbol{K}_{k,t}\boldsymbol{\Pi}_{\boldsymbol{u}_k,t})\boldsymbol{I}_r - 2\eta\boldsymbol{G}_{k-1,t}^\top\boldsymbol{G}_{k-1,t})\boldsymbol{u}_{k,t}^\top \\
&\quad + 2\eta\boldsymbol{R}_{k,t}' + \eta^2\boldsymbol{R}_{k,t},
\end{aligned}
$$

where $\boldsymbol{R}_{k,t}$ and $\boldsymbol{R}_{k,t}'$ are non-negative real numbers given by

$$\boldsymbol{R}_{k,t} = \boldsymbol{u}_{k,t}(\lambda_k\boldsymbol{I}_r - \boldsymbol{X}_t^\top\boldsymbol{X}_t)^2\boldsymbol{u}_{k,t}^\top,$$
$$\boldsymbol{R}_{k,t}' = \boldsymbol{u}_{k,t}(\sigma_1^2(\boldsymbol{K}_{k,t}\boldsymbol{\Pi}_{u_k,t})\boldsymbol{I}_r - \boldsymbol{\Pi}_{\boldsymbol{u}_k,t}\boldsymbol{K}_{k,t}^\top\boldsymbol{K}_{k,t}\boldsymbol{\Pi}_{\boldsymbol{u}_k,t})\boldsymbol{u}_{k,t}^\top.$$

It then follows that

$$\sigma_1^2(\boldsymbol{u}_{k,t+1}) \geq (1 + 2\eta\lambda_k - 2\eta\sigma_1^2(\boldsymbol{u}_{k,t}) - 2\eta\sigma_1^2(\boldsymbol{K}_{k,t}\boldsymbol{\Pi}_{\boldsymbol{u}_k,t})) \cdot \sigma_1^2(\boldsymbol{u}_{k,t}) - 2\eta\sigma_1^2(\boldsymbol{u}_{k,t}\boldsymbol{G}_{k-1,t}^\top),$$

which concludes the proof. ∎

Lemma 25 provides an upper bound on $\sigma_1(\boldsymbol{u}_{k,t+1}\boldsymbol{K}_{k,t+1}^\top)$ in terms of the preceding iteration.

**Lemma 25** *Suppose $\eta \leq \frac{1}{12\lambda_1}$ and $\sigma_1^2(\boldsymbol{X}_t) \leq 2\lambda_1$. For any $k \leq r$, if $\sigma_1^2(\boldsymbol{u}_{k,t}) > 0$, then we have*

$$
\begin{aligned}
&\sigma_1(\boldsymbol{u}_{k,t+1}\boldsymbol{K}_{k,t+1}^\top) \\
&\leq \big(1 + \eta\lambda_k + \eta\lambda_{k+1} - 2\eta\sigma_1^2(\boldsymbol{u}_{k,t}) - 2\eta\sigma_1^2(\boldsymbol{K}_{k,t}\boldsymbol{\Pi}_{\boldsymbol{u}_k,t}) + 25\eta^2\lambda_1^2\big) \cdot \sigma_1(\boldsymbol{u}_{k,t}\boldsymbol{K}_{k,t}^\top) \\
&\quad + 3\eta\sigma_1(\boldsymbol{u}_{k,t}\boldsymbol{G}_{k-1,t}^\top)\sigma_1(\boldsymbol{K}_{k,t}\boldsymbol{G}_{k-1,t}^\top).
\end{aligned}
$$

**Proof** Substituting (24) and (25) into $\boldsymbol{u}_{k,t+1}\boldsymbol{K}_{k,t+1}^\top$ gives that

$$
\begin{aligned}
\boldsymbol{u}_{k,t+1}\boldsymbol{K}_{k,t+1}^\top &= \boldsymbol{u}_{k,t}\boldsymbol{K}_{k,t}^\top + \eta\lambda_k\boldsymbol{u}_{k,t}\boldsymbol{K}_{k,t}^\top + \eta\boldsymbol{u}_{k,t}\boldsymbol{K}_{k,t}^\top\boldsymbol{\Gamma}_k - 2\eta\boldsymbol{u}_{k,t}\boldsymbol{X}_t^\top\boldsymbol{X}_t\boldsymbol{K}_{k,t}^\top + \eta^2\boldsymbol{E}, \\
&= \boldsymbol{B} + \boldsymbol{C} - 2\eta\boldsymbol{D} + \eta^2\boldsymbol{E},
\end{aligned}
$$

where

$$\boldsymbol{B} = \boldsymbol{u}_{k,t}\boldsymbol{K}_{k,t}^\top\left(\frac{1}{2}\boldsymbol{I}_{d-k} - 2\eta\boldsymbol{K}_{k,t}\boldsymbol{\Pi}_{\boldsymbol{u}_{k,t}}\boldsymbol{K}_{k,t}^\top\right)$$

$$\boldsymbol{C} = \boldsymbol{u}_{k,t}\boldsymbol{K}_{k,t}^\top\left(\frac{1}{2}\boldsymbol{I}_{d-k} + \eta\lambda_k\boldsymbol{I}_{d-k} - 2\eta\sigma_1^2(\boldsymbol{u}_{k,t})\boldsymbol{I}_{d-k} + \eta\boldsymbol{\Gamma}_k - 2\eta\boldsymbol{K}_{k,t}(\boldsymbol{I}_r - \boldsymbol{\Pi}_{\boldsymbol{u}_k,t})\boldsymbol{K}_{k,t}^\top\right),$$

$$\boldsymbol{D} = \boldsymbol{u}_{k,t}\boldsymbol{G}_{k-1,t}^\top\boldsymbol{G}_{k-1,t}\boldsymbol{K}_{k,t}^\top,$$

$$\boldsymbol{E} = \lambda_k\boldsymbol{u}_{k,t}\boldsymbol{K}_{k,t}^\top\boldsymbol{\Gamma}_k - \boldsymbol{u}_{k,t}\boldsymbol{X}_t^\top\boldsymbol{X}_t\boldsymbol{K}_{k,t}^\top\boldsymbol{\Gamma}_k - \lambda_k\boldsymbol{u}_{k,t}\boldsymbol{X}_t^\top\boldsymbol{X}_t\boldsymbol{K}_{k,t}^\top + \boldsymbol{u}_{k,t}(\boldsymbol{X}_t^\top\boldsymbol{X}_t)^2\boldsymbol{K}_{k,t}^\top.$$

By the singular value inequality,

$$\sigma_1(\boldsymbol{u}_{k,t+1}\boldsymbol{K}_{k,t+1}^\top) \le \sigma_1(\boldsymbol{B}) + \sigma_1(\boldsymbol{C}) + 2\eta\sigma_1(\boldsymbol{D}) + \eta^2\sigma_1(\boldsymbol{E}).$$

For the first term $\boldsymbol{B}$, observe that

$$
\begin{aligned}
(\boldsymbol{u}_{k,t}\boldsymbol{u}_{k,t}^\top)^{-1/2}\boldsymbol{B} &= (\boldsymbol{u}_{k,t}\boldsymbol{u}_{k,t}^\top)^{-1/2}\boldsymbol{u}_{k,t}\boldsymbol{K}_{k,t}^\top\left(\frac{1}{2}\boldsymbol{I}_{d-k} - \boldsymbol{K}_{k,t}\boldsymbol{\Pi}_{\boldsymbol{u}_k,t}\boldsymbol{K}_{k,t}^\top\right) \\
&= \left(1/2 - \sigma_1^2((\boldsymbol{u}_{k,t}\boldsymbol{u}_{k,t}^\top)^{-1/2}\boldsymbol{u}_{k,t}\boldsymbol{K}_{k,t}^\top\right) \cdot (\boldsymbol{u}_{k,t}\boldsymbol{u}_{k,t}^\top)^{-1/2}\boldsymbol{u}_{k,t}\boldsymbol{K}_{k,t}^\top \\
&= \left(1/2 - \sigma_1^2(\boldsymbol{K}_{k,t}\boldsymbol{\Pi}_{\boldsymbol{u}_k,t})\right) \cdot (\boldsymbol{u}_{k,t}\boldsymbol{u}_{k,t}^\top)^{-1/2}\boldsymbol{u}_{k,t}\boldsymbol{K}_{k,t}^\top.
\end{aligned}
$$

where we use the equality $\sigma_1(\boldsymbol{K}_{k,t}\boldsymbol{\Pi}_{\boldsymbol{u}_k,t}) = \sigma_1((\boldsymbol{u}_{k,t}\boldsymbol{u}_{k,t}^\top)^{-1/2}\boldsymbol{u}_{k,t}\boldsymbol{K}_{k,t}^\top)$. Thus,

$$\sigma_1(\boldsymbol{B}) = \left(1/2 - \sigma_1^2(\boldsymbol{K}_{k,t}\boldsymbol{\Pi}_{\boldsymbol{u}_k,t})\right) \cdot \sigma_1(\boldsymbol{u}_{k,t}\boldsymbol{K}_{k,t}^\top).$$

For the second term $\boldsymbol{C}$, by the singular value inequality,

$$
\begin{aligned}
&\sigma_1(\boldsymbol{C}) \\
&\le \sigma_1\left(\frac{1}{2}\boldsymbol{I}_{d-k} + \eta\lambda_k\boldsymbol{I}_{d-k} - 2\eta\sigma_1^2(\boldsymbol{u}_{k,t})\boldsymbol{I}_{d-r} + \eta\boldsymbol{\Gamma}_k - 2\eta\boldsymbol{K}_{k,t}(\boldsymbol{I}_r - \boldsymbol{\Pi}_{\boldsymbol{u}_k,t})\boldsymbol{K}_{k,t}^\top\right) \cdot \sigma_1(\boldsymbol{u}_{k,t}\boldsymbol{K}_{k,t}^\top) \\
&\le \left(1/2 + \eta\lambda_k - 2\eta\sigma_1^2(\boldsymbol{u}_{k,t}) + \eta\lambda_{k+1}\right) \cdot \sigma_1(\boldsymbol{u}_{k,t}\boldsymbol{K}_{k,t}^\top).
\end{aligned}
$$

For the third term $\boldsymbol{D}$, $\sigma_1(\boldsymbol{D}) \le \sigma_1(\boldsymbol{u}_{k,t}\boldsymbol{G}_{k-1,t}^\top)\sigma_1(\boldsymbol{K}_{k,t}\boldsymbol{G}_{k-1,t}^\top)$. For the fourth term $\boldsymbol{E}$, since $\sigma_1^2(\boldsymbol{X}_t) \le 2\lambda_1$, we have

$$\sigma_1(\boldsymbol{E}) \le 25\lambda_1^2\sigma_1(\boldsymbol{u}_{k,t}\boldsymbol{K}_{k,t}) + 8\lambda_1\sigma_1(\boldsymbol{u}_{k,t}\boldsymbol{G}_{k-1,t}^\top)\sigma_1(\boldsymbol{K}_{k,t}\boldsymbol{G}_{k-1,t}^\top).$$

Combining, we prove the lemma. ∎

Lemma 26 provides an upper bound on a specific ratio, which is used in the proof of Lemma 22. It serves as a new variant of the SNR argument.

**Lemma 26** *Suppose* $\eta \le \frac{\Delta}{100\lambda_1^2}$, $\sigma_1^2(\boldsymbol{X}_t) \le 2\lambda_1$, *and* $\lambda_{k+1} \le \lambda_k - \Delta$. *Let*

$$\mathrm{ratio} := \frac{1 + \eta\lambda_k + \eta\lambda_{k+1} - 2\eta\sigma_1^2(\boldsymbol{u}_{k,t}) - 2\eta\sigma_1^2(\boldsymbol{K}_{k,t}\boldsymbol{\Pi}_{\boldsymbol{u}_k,t}) + 25\eta^2\lambda_1^2}{1 + 2\eta\lambda_k - \eta\Delta/4 - 2\eta\sigma_1^2(\boldsymbol{u}_{k,t}) - 2\eta\sigma_1^2(\boldsymbol{K}_{k,t}\boldsymbol{\Pi}_{\boldsymbol{u}_k,t})}.$$

*Then* $\mathrm{ratio} \le 1 - \eta\Delta/6$.

**Proof** Since $\eta \le \frac{\Delta}{100\lambda_1^2}$ and $\lambda_{k+1} < \lambda_k - \Delta$, we have

$$\mathrm{ratio} \le \frac{1 - \eta\Delta/4 + s_0}{1 + \eta\Delta/4 + s_0},$$

where

$$s_0 = 2\eta\lambda_k - \eta\Delta/2 - 2\eta\sigma_1^2(\boldsymbol{u}_{k,t}) - 2\eta\sigma_1^2(\boldsymbol{K}_{k,t}\boldsymbol{\Pi}_{k,t}) \in [-1/2, 1/2].$$

Since the function $g_9(s) = \frac{1 - \eta\Delta/4 + s}{1 + \eta\Delta/4 + s}$ is increasing on $[-1/2, 1/2]$, we have

$$\mathrm{ratio} \le \frac{1 - \eta\Delta/4 + 1/2}{1 + \eta\Delta/4 + 1/2} \le 1 - \eta\Delta/6,$$

which concludes the proof. ∎

# E ADDITIONAL EXPERIMENTS

In this section provide additional experiments to support and illustrate our theoretical results.

## E.1 RANK-TWO MATRIX APPROXIMATION

Our first extended experiment examines rank-two matrix approximation with varying dimension $d$ and initial magnitude $\varpi$. Specifically, we will choose $d$ from the set $\{1000, 2000, 4000\}$ and choose $\varpi$ from the set $\{0.001, 0.5, 2\}$. For each $d$, we set $\boldsymbol{\Sigma} = \mathrm{diag}(\boldsymbol{a}, \boldsymbol{e})$, where $\boldsymbol{a} \in \mathbb{R}^r$ is a decreasing arithmetic sequence starting from 1 to 0.5 and $\boldsymbol{e} \in \mathbb{R}^{d-r}$ is an arithmetic sequence transitioning from 0.3 to zero. Let $\boldsymbol{X}_0 = \varpi \boldsymbol{N}_0$ with the entries of $\boldsymbol{N}_0$ independently drawn from $\mathcal{N}(0, \frac{1}{d})$. We compute the GD sequence $\boldsymbol{X}_t$ with a step size of 0.1 and evaluate the errors $\|\boldsymbol{\Sigma}_r - \boldsymbol{X}_t\boldsymbol{X}_t^\top\|_\mathrm{F}$, where $\boldsymbol{\Sigma}_r = \mathrm{diag}(\boldsymbol{a}, \boldsymbol{0})$ is the best rank-r approximation to $\boldsymbol{\Sigma}$. The error curves of GD for different settings are displayed in Figure 2.

Figure 2 demonstrate that all the error curves exhibit the similar behaviors. The only differences lie on the first stage.

- When we use a small $\varpi = 0.001$, the error does not rapidly change at the beginning. This is because $\|\boldsymbol{X}_t\|_\mathrm{F}$ is close to zero and the error $\|\boldsymbol{\Sigma}_r - \boldsymbol{X}_t\boldsymbol{X}_t\|_\mathrm{F}$ is approximately $\|\boldsymbol{\Sigma}_r\|_\mathrm{F}$. This period of time corresponds to the second property of Theorem 6.

- When we use $\varpi = 2$, we find the error first drops rapidly from a large value to $\|\boldsymbol{\Sigma}_r\|$. This corresponds to the Lemma 8 and the first property in Theorem 5.

- When we use $\varpi = 0.5$, the first stage nearly disappears. This means that $T_{\boldsymbol{u}_1}$ in Theorem 5 is small, especially compared with the case where $\varpi = 0.001$.

In addition, we want to mention that if we use $\varpi = 10$ to initialize the algorithm and keep other settings unchanged, then the GD sequence will diverge. This serves as a supplementary to the above experimental results.

## E.2 GENERAL RANK MATRIX APPROXIMATION

Our second experiment examines general rank matrix approximation, where we fix dimension $d = 1000$ and vary the rank $r$ across $\{2, 6, 10\}$. In addition, for each setting, we examine different initial magnitudes $\varpi \in \{0.001, 0.5, 2\}$. Our setting for $\boldsymbol{\Sigma}$ is the same as before, that is, $\boldsymbol{\Sigma} = \mathrm{diag}(\boldsymbol{a}, \boldsymbol{e})$ with $\boldsymbol{a} \in \mathbb{R}^r$ and $\boldsymbol{e}^{d-r}$ being two arithmetic sequences. We initialize GD using $\boldsymbol{x}_0 = \varpi \boldsymbol{N}_0$ and we compute the GD sequence and the errors $\|\boldsymbol{\Sigma}_r - \boldsymbol{X}_t\boldsymbol{X}_t\|_\mathrm{F}$. The results are displayed in Figure 3.

As the results demonstrate, the effects of $\varpi$ is similar to the one in Section E.1. Moreover, we observe another interesting phenomenon that may need additional explanations. Figure 3 shows that the error curve for larger rank $r$ is smoother than the one for smaller rank $r$. Our explanation is that for larger rank $r$, the differences between successive eigenvalues are smaller. Thus, it is harder to distinguish the associated eigenvectors, and all the eigenvectors may be learned together. As a result, the error curve remains decreasing along the iterations.

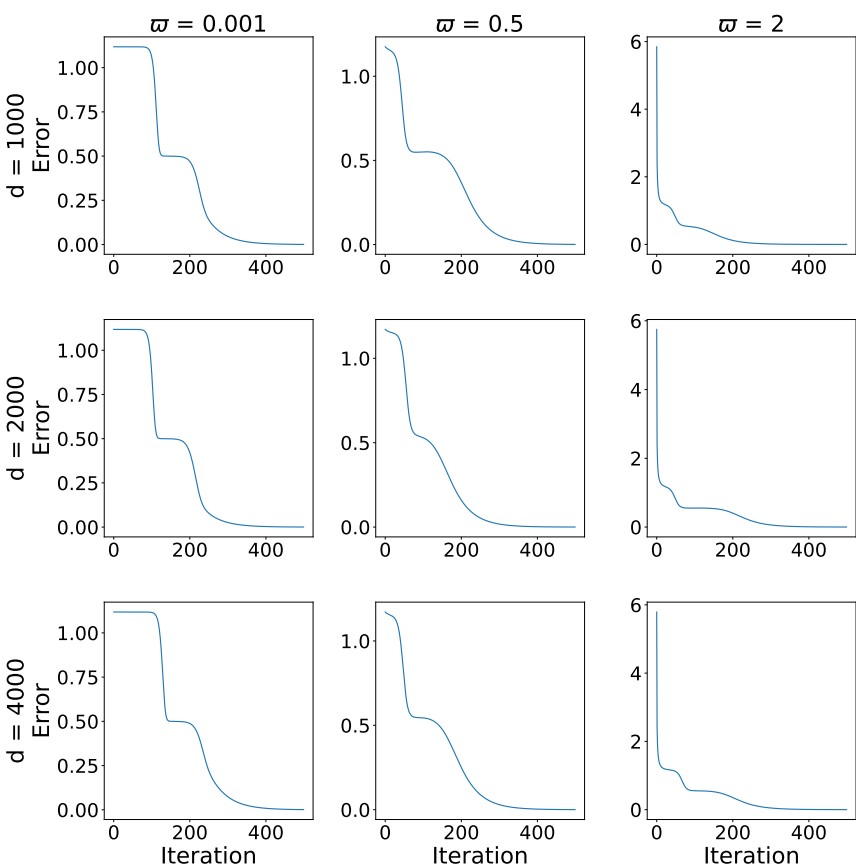

Figure 2: Error curves of GD, measured by $\|\boldsymbol{\Sigma}_r - \boldsymbol{X}_t \boldsymbol{X}_t^\top\|_{\mathrm{F}}$, for rank-two matrix approximation. The columns represent different initial magnitudes $\varpi = 0.001, 0.5, 2$. The rows represent different dimensions $d = 1000, 2000, 4000$.

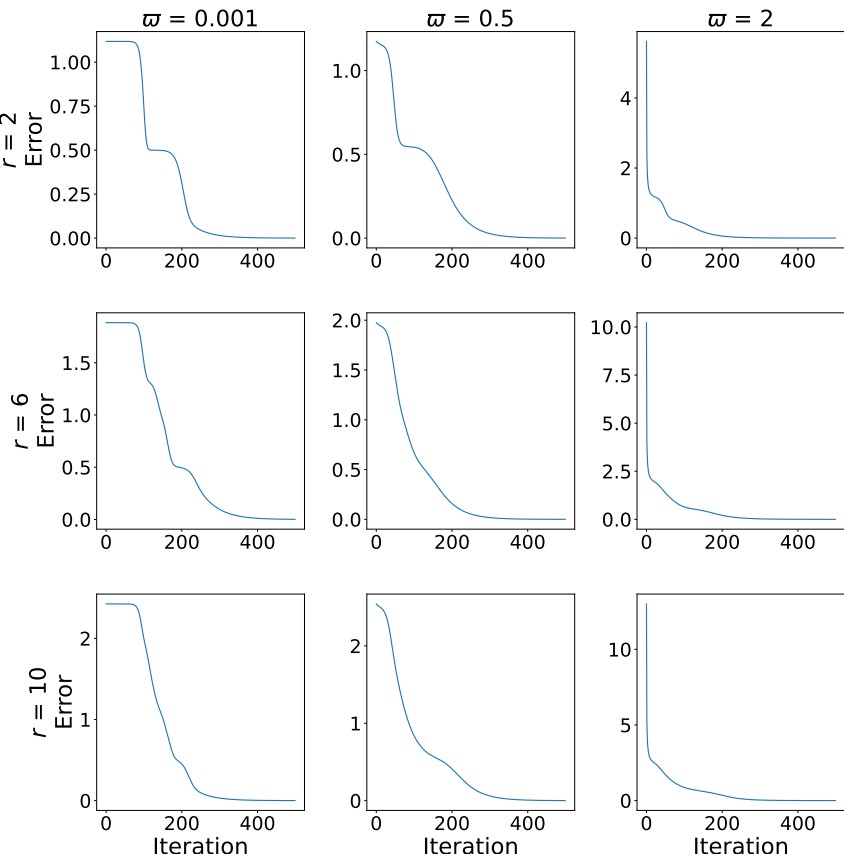

Figure 3: Error curves of GD, measured by $\|\boldsymbol{\Sigma}_r - \boldsymbol{X}_t\boldsymbol{X}_t^\top\|_{\mathrm{F}}$, for general rank matrix approximation. The dimension $d$ is set as 1000. Different rows represent different rank $r$. Different columns represent different initial magnitudes $\varpi$.