# OpenReview forum: "Gradient descent in matrix factorization: Understanding large initialization"
_auai.org/UAI/2024/Conference — UAI 2024 poster_

### Official Review · Reviewer_tusM · 2024-03-13

**Q2-1 Originality-Novelty:** 2
**Q2-2 Correctness-Technical Quality:** 3
**Q2-5 Clarity Of Writing:** 2

**Q1 Summary And Contributions:**

This work proves the convergence of symmetric matrix factorization under a large initialization

**Q2-3 Extent To Which Claims Are Supported By Evidence:**

3: Good: the main claims are supported by convincing evidence (in the form of adequate experimental evaluation, proofs, (pseudo-)code, references, assumptions).

**Q2-4 Reproducibility:**

3: Good: key resources (e.g. proofs, code, data) are available and key details (e.g. proofs, experimental setup) are sufficiently well-described for competent researchers to confidently reproduce the main results.

**Q3 Main Strengths:**

The paper writing is clear. The problem of matrix factorization is fundamental and important. The technical aspect is solid.

The problem of starting from a large initialization is both important and interesting

**Q4 Main Weakness:**

I think the main weaknesses are:

1. Unclear main result. The main result in theorem 6 hardly admits much understanding. A little explanation is given to the item 4 of the theorem, but I find it more speculative than justified

2. The interpretation of item (4) of theorem 6. The authors claims that this explains the incremental learning phenomenon, but I do not think so. Item (6) only shows what happens after time $t_k$, but for incremental learning to happen, nothing should change much after $t_k$, and this aspect seems missing. Also, I think it is fair to give this work a reference for incremental learning: https://arxiv.org/abs/1312.6120

3. Contribution sufficiency. I am not sure if the contribution is sufficient. Given that the cleanest contribution: that GD converges under a large init follows rather simply from previous results

**Q5 Detailed Comments To The Authors:**

I have a few questions and suggestions:

1. What is a "population version" of the matrix sensing problem? What is its difference with the standard version?

2. Why is it without loss of generality to assume that $\Sigma$ is diagonal?

3. Why does the theory depends crucially on the gap $\Delta$? This is interesting and I think the authors ought to have an extensive discussion of this in the draft

4. On the constructive side, I think it would be nice for the authors to mention the relevance of their theory to self-supervised learning, for example, see https://arxiv.org/abs/2303.15438. It is also nice to compare with the case of stochastic gradient descent, where convergence to the saddle is likely at a high SNR: https://arxiv.org/abs/2107.11774

**Q9 Complying With Reviewing Instructions:**

Yes

---

> ### Author Rebuttal · Authors · 2024-04-06
>
> ## Rebuttal - 1
>
> We would like to thank Reviewer tusM for the constructive review. In this rebuttal, we will address your concerns in detail.
>
> - *Unclear main result. The main result in theorem 6 hardly admits much understanding. A little explanation is given to the item 4 of the theorem, but I find it more speculative than justified*
> - Our Theorem 6 has two main results:
>     - First, it establishes that GD achieves $\epsilon$ accuracy in $t_{\mathcal{R}}+O(\log(1/\epsilon))$ iterations. In the expression of $t_{\mathcal{R}}$, all the quantities are **logarithmic terms** except $\{t^*_k\}$. The quantity $t_k^*$  characterizes the time when the $(k+1)$-th signal strength is no longer smaller than a geometrically decaying sequence, which is an informative characterization.
>     - Second, it establishes the incremental learning phenomenon, aka Property 4. It shows that when $t$ is larger than $t_k$, both $\sigma_1(u_{k,t}K_{k,t}^\top)$  and $|p_{k,t}|$ converge **linearly** to zero. Therefore, for large $t$, these two quantities are approximately zero, and thus remain almost unchanged. In addition, by recalling the definition, $p_{k,t}=0$ means that $\sigma_1^2(u_{k,t})=\lambda_k$. Thus, for sufficiently large $t$, $\sigma_1^2(u_{k,t})$ is approximately $\lambda_k$ and remains almost unchanged.
>
>     In the revised version, we will improve the presentation of these results.
>
> - *The interpretation of item (4) of theorem 6. The authors claims that this explains the incremental learning phenomenon, but I do not think so. Item (6) only shows what happens after time $t_k$, but for incremental learning to happen, nothing should change much after $t_k$, and this aspect seems missing. Also, I think it is fair to give this work a reference for incremental learning: https://arxiv.org/abs/1312.6120*
> - We think our answer to your previous question has addressed this question. In particular, the item (4) does prove that ($\sigma_1^2(u_{k,t})$, $\sigma_1(u_{k,t}K_{k,t}^\top)$ ) do not change much after $t_k$, or more precisely $t_k$ + a reasonably large time. This explains the incremental learning phenomenon. We will also discuss the recommended reference in our revised version. Thanks for the recommendation.
> - *Contribution sufficiency. I am not sure if the contribution is sufficient. Given that the cleanest contribution: that GD converges under a large init follows rather simply from previous results*
> - In this paper, we develop a novel signal-to-noise analytic framework to address the problem of GD with large initialization, which does not receive any study before. While being a little incomplete, the results in our paper can greatly enhance our understanding on this problem, such as the incremental learning phenomenon. Also, we believe brand new theoretical tools, such as dynamic systems, are needed to fully address the problem. Thus, we would like to leave such investigation to future work.

---

### Official Review · Reviewer_siQA · 2024-03-15

**Q2-1 Originality-Novelty:** 3
**Q2-2 Correctness-Technical Quality:** 3
**Q2-5 Clarity Of Writing:** 3

**Q1 Summary And Contributions:**

Authors analyse gradient descent (GD) applied to the problem
	min_X ||Sigma - XX^T||_F^2
where Sigma is positive semidefinite (PSD). GD convergence will depend on the initialization, X_0, and authors analyse the case when X_0 is "large", which fills a gap in the literature. The paper is overall well written, although the introduction should be improved and the motivations much better explained.

**Q2-3 Extent To Which Claims Are Supported By Evidence:**

2: Fair: the main claims are somewhat supported by evidence (but the experimental evaluation may be weak, or does not match entirely with the claims, important baselines may be missing, proofs contain important ideas but lack rigor, algorithmic details are only discussed superficially, references are imprecise, assumptions are not sufficiently motivated or explicated, etc.).

**Q2-4 Reproducibility:**

3: Good: key resources (e.g. proofs, code, data) are available and key details (e.g. proofs, experimental setup) are sufficiently well-described for competent researchers to confidently reproduce the main results.

**Q3 Main Strengths:**

The paper provides a theoretical analysis of GD on the problem min_X ||Sigma - XX^T||_F^2, when the initial points is "large", which was not done before in the literature.

**Q4 Main Weakness:**

- The important notions used in the paper should be defined earlier. For example, authors talk about large/small/random/benign initializations in the first page, mentioning previous works, but never defining what these mean exactly. This is confusing for readers not familiar with these notions (like me). Also the "eigengap" is mentioned but only defined much later. (In fact, the definition used is not standard so it would be good to give it right away.)
- I think authors should better emphasize why large initialization is interesting to analyse. They mention on page 5 that "practitioners often use large initializations" but why? Does it behave better in practice? leading to faster convergence? This is never discussed, and I believe crucial otherwise we do not really see the point of the paper. In fact, authors could provide some numerical experiments to back up this discussion.
- In Theorem 6, the step size, $\eta$, could be rather small. It might be good to discuss this, and what would be recomended in practice. Moreover, how does it compare to stepsizes used in the case of small initializations from previous works? If one can use (much) larger stepsizes in the case of small initializations, then large initializations might not be that practical/useful. A discussion would be helpful, and maybe some illustrative numerical experiments.

**Q5 Detailed Comments To The Authors:**

Some other comments:
- Please clarify "the behavior of GD in specific time"
- Theorem 6: ... Then we have: we have ... we have ... please avoid using so many "we have" (in fact, Nick Higham in his book on math writing recommends using as few as possible).
- "rather than one times"

**Q9 Complying With Reviewing Instructions:**

Yes

---

> ### Author Rebuttal · Authors · 2024-04-06
>
> We would like to thank Reviewer siQA for the constructive review. In the following, we will address all of your concerns in detail.
>
> - *The important notions used in the paper should be defined earlier. For example, authors talk about large/small/random/benign initializations in the first page, mentioning previous works, but never defining what these mean exactly.*
> - Except for large initialization, all other three notions of initialization have been used in previous literature. Here let us give a precise definition:
>     - Benign initialization: the initial point is located in a region surrounding the global minimum, such as $\mathcal{R}$ in (11) in Section 3.
>     - Random initialization: use random mechanism to initialize the gradient descent. This is generally far from the global minimum, or the benign region $\mathcal{R}$. For example, one may initialize $X_0=\varpi N_0\in\mathbb{R}^{d\times r},$ where $N_0$ has i.i.d. $N(0,1/d)$ entries and $\varpi$ is the order of initialization, as we mentioned in Page 1.
>     - Small initialization: the norm of the initial point $|X_0|_F$  tends to zero as $d$ tends to infinity. When using the random initialization $X_0=\varpi N_0,$ the norm $\|X_0\|_F$ and $\varpi$ share the same order. Thus, small initialization means that $\varpi$ tends to zero as $d$ tends to infinity.
>     - Large initialization: the norm of the initial point $\|X_0\|_{F}$  is of constant order even when $d$ tends to infinity. When using the random initialization $X_0=\varpi N_0,$ large initialization means $\varpi$ is of constant order as $d$ tends to infinity. (Note: here we only require $\varpi=\Theta(1)$, since if $\varpi$ is larger than $O(1)$, then GD may simply diverge.)
>     - One more note: previous literature typically requires $\varpi$ to decay exponentially fast as $d$ tends to infinity. For example, $\varpi\lesssim d^{-3\kappa^2},$ where $\kappa$ is the condition number. This clearly belongs to the small initialization regime.
> - *Also the "eigengap" is mentioned but only defined much later. (In fact, the definition used is not standard so it would be good to give it right away.)*
> - In our paper, the eigengap $\Delta$ refers to (1) the difference between $r$-th and $(r+1)$-th largest eigenvalues in Section 2, 3 and 4.1; (2) the smallest gap between the first $(r+1)$ eigenvalues in Section 4.2, where we assume the top $(r+1)$ eigenvalues are strictly decreasing. The decreasing eigenvalue assumption is only needed in Section 4.2 because of our inductive argument. In our revised paper, we would make suitable revisions to improve the readability.
> - *I think authors should better emphasize why large initialization is interesting to analyse.*
> - Let us explain this point using a numerical experiment. *We have prepared a figure but we currently have trouble uploading it.* Before we figure out how to upload this experimental results, we provide the experimental setting in the summary of the rebuttal. Based on this, we believe the reviewer can easily implement the algorithm and obtain the results.
> - *In Theorem 6, the step size $\eta$ could be rather small. It might be good to discuss this, and what would be recommended in practice. Moreover, how does it compare to stepsizes used in the case of small initializations from previous works? If one can use (much) larger stepsizes in the case of small initializations, then large initializations might not be that practical/useful. A discussion would be helpful, and maybe some illustrative numerical experiments.*
> - First of all, the step size is a constant in our work and related works, which does not depend on the dimension $d$. Second, to achieve the local linear convergence in Theorem 2, the condition $\eta\lesssim \Delta^2/\lambda_1^3$ is typically required. This is also required in related works too. Third, to derive Theorem 6, we require the eigengap $\Delta$ to be the smallest eigengaps between the top $r+1$ eigenvalues. This is distinct from previous literature and it is indeed a weak point. However, when all the top $r+1$ eigenvalues are well-separated, then this weakness is mitigated. Finally, optimizing the condition of $\eta$ is possible in certain specific regime (such as the rank-1 case), but we do not present them because it is not the main focus of the paper.
> - *Please clarify "the behavior of GD in specific time”*
> - Our trajectory analysis establishes (1) upper bounds on $t_{{\rm init},1},$ $T_{u_k},$ (2) properties of $\sigma_1^2(u_{k,t})$ after $t\geq t_{{\rm init},k}+T_{u_k}$; properties of $|p_{k,t}|$ after $t_k$; and properties of $X_t$ after $t_{\mathcal{R}}$. We will revise the presentation of Theorem 6 and the previous paragraphs to further improve the readability.
> - *Theorem 6: ... Then we have: we have ... we have ... please avoid using so many "we have" (in fact, Nick Higham in his book on math writing recommends using as few as possible)*
> - Yes! We will take this into consideration when we improve the presentation.

---

### Official Review · Reviewer_LS35 · 2024-03-23

**Q2-1 Originality-Novelty:** 2
**Q2-2 Correctness-Technical Quality:** 3
**Q2-5 Clarity Of Writing:** 4

**Q1 Summary And Contributions:**

The submitted draft discusses gradient descent in matrix factorization with large initialization  and presents a novel analysis of the algorithm's behavior in optimization problems. By leveraging Signal-to-Noise Ratio (SNR) arguments, the study offers insights into the trajectory and convergence properties of gradient descent in scenarios with large initialization. it makes nontrivial  theoretical contributions to the field of low-rank optimization and implicit regularization, providing practical implications for optimizing matrix factorization problems.

**Q2-3 Extent To Which Claims Are Supported By Evidence:**

3: Good: the main claims are supported by convincing evidence (in the form of adequate experimental evaluation, proofs, (pseudo-)code, references, assumptions).

**Q2-4 Reproducibility:**

3: Good: key resources (e.g. proofs, code, data) are available and key details (e.g. proofs, experimental setup) are sufficiently well-described for competent researchers to confidently reproduce the main results.

**Q3 Main Strengths:**

The paper is largely well-written, the results are clearly stated and presented. The paper makes non-tirvial theoretical contributions by leveraging Signal-to-Noise Ratio (SNR) arguments to analyze the behavior of gradient descent in matrix factorization problems The theoretical analysis is rigorous and satisifying to read and understand, I enjoyed the paper mostly though I could not go through all the proofs (it is a dense paper), but I mostly believe the results.

I especially like the limitations included in the concluding remarks, which clearly talks about possible future directions to build this work on.

**Q4 Main Weakness:**

Applicability: I am concerned about how interested the UAI community would be in this narrow/niche problem. If the authors could expand the discussion on motivating the problem with some real world examples, and discuss the importance/shortcomings of other competing initialization methods, that would be helpful to getting a better understanding. While I understand the importance of implicit regularization and how this paper adds to that discussion, I feel the overall contribution is narrow/too niche compared to other papers submitted at this conference.

Some parts of the paper are dense, but that is the nature of hte problem and the analysis.

**Q5 Detailed Comments To The Authors:**

-

**Q9 Complying With Reviewing Instructions:**

Yes

---

> ### Author Rebuttal · Authors · 2024-04-06
>
> We would like to express our gratitude to Reviewer LS35 for the insightful review. In this rebuttal, we will address all of your concerns in detail.
>
> - *I am concerned about how interested the UAI community would be in this narrow/niche problem. If the authors could expand the discussion on motivating the problem with some real world examples, and discuss the importance/shortcomings of other competing initialization methods, that would be helpful to getting a better understanding*
> - There are several real-world problems motivating our studied problems.
>     - First, our studied problem is a population version of the matrix sensing problem, which is useful in compressed sensing. Specifically, suppose $\Sigma\in\mathbb{R}^{d\times d}$ is a rank-$r$ unknown symmetric positive semidefinite matrix that we aim to recover. Let $A_1,\ldots,A_m$ be $m$ given symmetric sensing matrices in $\mathbb{R}^{d\times d}$ and $y_1,\ldots,y_m$ be the linear measurements:
>
>         $$
>         y_i=\left<A_i,\Sigma\right>,\quad i=1,\ldots,m.
>         $$
>
>         The matrix sensing problem aims to recover $\Sigma$ from the data $\{(y_i,A_i)\}_{i=1}^m$, by solving the following least square minimizing problem:
>
>         $$
>         \hat\Sigma=\hat X\hat X^\top,\quad \hat X=\arg\min_{X\in\mathbb{R}^{d\times r}}f(X)\overset{\rm def}{=}\frac{1}{4m}\sum_{i=1}^m(y_i-\left<A_i,XX^\top\right>)^2.
>         $$
>
>         Starting from an initial point $X_0\in\mathbb{R}^{d\times r}$, the gradient descent algorithm will update $X_t$ as follows:
>
>         $$
>         X_t=X_{t-1}-\eta \nabla f(X_{t-1}),
>         $$
>
>         where $\nabla f(X)$ is the gradient of $f$ given by
>
>         $$
>         \nabla f(X)=-\frac{1}{m}\sum_{i=1}^m(y_i-\left<A_i,XX^\top\right>)A_iX=-\frac{1}{m}\sum_{i=1}^m\left<A_i,\Sigma-XX^\top\right>\cdot A_iX.
>         $$
>
>         In most matrix sensing literature, a standard assumption is that $\{A_i\}_{i=1}^m$ are i.i.d. with independent $N(0,1)$ entries. Under this assumption, the gradient $\nabla f(X)$ is approximately
>
>         $$
>         \nabla f(X)\approx\mathbb{E}\nabla f(X)=-(\Sigma-XX^\top)X
>         $$
>
>         by the law of large numbers (or more sophisticated concentration properties established in [1]). Consequently, the population version of the gradient descent reduces to
>
>         $$
>         X_t=X_{t-1}+\eta(\Sigma-X_tX_t^\top)X_t,
>         $$
>
>         which is the gradient descent (10) in our paper. Many works build on the intuition that for sufficiently large $m$, matrix sensing is approximately the matrix factorization, e.g., [1,2,3,4].
>
>         [1] **[Guaranteed minimum-rank solutions of linear matrix equations via nuclear norm minimization](https://epubs.siam.org/doi/abs/10.1137/070697835)**
>
>         [2] **[The global optimization geometry of low-rank matrix optimization](https://ieeexplore.ieee.org/abstract/document/9314092/)**
>
>         [3] **[Global optimality in low-rank matrix optimization](https://ieeexplore.ieee.org/abstract/document/8357489/)**
>
>         [4] **[Algorithmic regularization in over-parameterized matrix sensing and neural networks with quadratic activations](https://proceedings.mlr.press/v75/li18a.html)**
>
>     - Second, the matrix factorization problem is the population version of the phase retrieval problem. This is similar to the matrix sensing problem, so we do not explain it in detail. See [5,6] for references.
>
>
>         [5] **[Gradient descent with random initialization: Fast global convergence for nonconvex phase retrieval](https://link.springer.com/article/10.1007/s10107-019-01363-6)**
>
>         [6] **[Implicit regularization in nonconvex statistical estimation: Gradient descent converges linearly for phase retrieval, matrix completion, and blind deconvolution](https://proceedings.mlr.press/v80/ma18c.html)**
>
> - *While I understand the importance of implicit regularization and how this paper adds to that discussion, I feel the overall contribution is narrow/too niche compared to other papers submitted at this conference.*
> - Our initial intention is to study the matrix sensing problem, which is practically meaningful. However, due to the technical challenges, we decided to divide this study into several stages. This paper examines the first stage and the simplest case: matrix factorization. We hope through such a study, we can (1) draw people’s attention to the problem of large initialization, whose practical meanings are explained in our response to the Reviewer siQA; (2) provide some useful analytic tools.
> - *Some parts of the paper are dense, but that is the nature of the problem and the analysis.*
> - We plan to further improve the presentation of this paper. For example, we will further clean the results and notations in Theorem 6.

---

### Meta-Review · Area_Chair_zc3K · 2024-04-15

Summary: The paper under consideration analyzes the behavior of gradient descent optimization for matrix factorization problems when initialized with large values. It introduces a novel theoretical framework based on Signal-to-Noise Ratio (SNR) concepts to examine the trajectory and convergence characteristics of the gradient descent algorithm in these high initialization scenarios. Through this analysis, the study makes original theoretical contributions to the fields of low-rank optimization and implicit regularization. The insights gained from this work offer practical implications for effectively optimizing matrix factorization problems by providing a deeper understanding of how gradient descent performs under large initialization conditions.

Meta-review: Based on the initial reviews, the verdict is that the authors have crafted an above-the-bar paper that presents a compelling proposal backed by solid reasoning. There is one pending borderline reject, but I believe the authors have handled these issues sufficiently. Based on the interaction between authors and reviewers, most concerns have been raised, and all the reviewers place the paper above the acceptance borderline. The authors have adequately answered and interacted with the authors' questions.